# Multitrait genome-wide analyses identify new susceptibility loci and candidate drugs to primary sclerosing cholangitis

Younghun Han [1,2,27], Jinyoung Byun [1,2,3,27], Catherine Zhu [1], Ryan Sun[4], Julia Y. Roh[5], Heather J. Cordell [6], Hyun-Sung Lee [7], Vikram R. Shaw[1], Sung Wook Kang[7], Javad Razjouyan[8,9,10,11], Matthew A. Cooley[12], Manal M. Hassan[13], Katherine A. Siminovitch[14,15], Trine Folseraas[16], David Ellinghaus [17], Annika Bergquist[18], Simon M. Rushbrook[19,20], Andre Franke[17], Tom H. Karlsen [21], Konstantinos N. Lazaridis [22], The International PSC Study Group*, Katherine A. McGlynn[23], Lewis R. Roberts [21] & Christopher I. Amos [1,2,3] ✉

Primary sclerosing cholangitis (PSC) is a rare autoimmune bile duct disease that is strongly associated with immune-mediated disorders. In this study, we implemented multitrait joint analyses to genome-wide association summary statistics of PSC and numerous clinical and epidemiological traits to estimate the genetic contribution of each trait and genetic correlations between traits and to identify new lead PSC risk-associated loci. We identified seven new loci that have not been previously reported and one new independent lead variant in the previously reported locus. Functional annotation and fine-mapping nominated several potential susceptibility genes such as *MANBA* and *IRF5*. Network-based in silico drug efficacy screening provided candidate agents for further study of pharmacological effect in PSC.

Primary sclerosing cholangitis (PSC) is a chronic, progressive autoimmune disorder of the bile duct[1–3]. Individuals with PSC are at risk of severe liver problems including a lifetime risk of cholangiocarcinoma of between 5 and 20%[4]. PSC is often associated with inflammatory bowel disease (IBD). Approximately 75% of individuals with PSC have IBD[2], most commonly ulcerative colitis (UC). Individuals with PSC are also more likely than those without PSC to have other autoimmune diseases, including type 1 diabetes, celiac disease, and thyroid disease. The shared etiology and underlying characteristics of these immune-mediated disorders remain incompletely understood.

Recent genome-wide association studies (GWAS) have identified ~19 loci associated with PSC among individuals of European ancestry[2,5]. Association analysis using the Immunochip genotype array data that specifically targeted known autoimmune-related disease regions identified three additional loci influencing PSC risk[6]. The development of PSC can be attributed to a combination of genetic and environmental factors[7]. Individuals with a family history of PSC have an increased risk of developing PSC suggesting that genetic influences play a critical role in susceptibility, which may act in concert with exposure to specific environmental factors. However, the genetic and environmental risk factors are not fully elucidated. As PSC is strongly associated with IBD[2], examining two traits together may provide better genetic insight into a common genetic etiology[8–11]. Few studies have been conducted to understand the shared genetic underpinning between PSC and other associated medical conditions.

Leveraging publicly available GWAS summary-level data[12–14] (Supplementary Data 1, "Methods"), we conducted cross-trait linkage disequilibrium (LD) score regression (LDSR) analysis[15,16] to determine whether there was a shared genetic contribution between polygenic phenotypes for multiple diseases and traits. We explored the directionality and degree of these relationships, and whether the genetic architecture between two traits is correlated or inversely correlated[17].

A full list of affiliations appears at the end of the paper. *A list of authors and their affiliations appears at the end of the paper. ✉e-mail: Chris.Amos@bcm.edu

We took advantage of the genetic overlap between traits to identify additional independent genetic variants for PSC alongside five immune-mediated disorders (Supplementary Data 2), highly correlated with PSC: Crohn's disease[18] (CD), UC[18], IBD[18], lupus[19], and primary biliary cirrhosis[20] (PBC) using multitrait analysis of GWAS[21] (MTAG). Although IBD is the umbrella term that includes CD and UC, we also surveyed the pairwise genetic correlation of PSC for CD and UC, respectively. We then performed functional fine-mapping analyses on the newly identified loci to elucidate potential functional characterization and biological mechanisms affecting PSC susceptibility. Since there is no medication proven to be effective for PSC treatment, we conducted network-based drug–disease proximity analysis to identify potential agents suitable for repurposing to PSC from the previously reported[13] and newly identified candidate genes in this study.

## Results

### PSC shows the shared genetic contributions among numerous clinical and epidemiological traits

We investigated the proportion of phenotypic variance explained by all common single-nucleotide polymorphisms (SNPs) for 134 clinical and epidemiological traits to identify potential comorbid conditions and to uncover traits that are causally involved in clinical course and epidemiologic associations using LDSR ("Methods"). We identified numerous traits showing moderate SNP-heritability in the observed scale (h2). The study workflow shown in Fig. 1 summarizes the steps from data preparation to subsequent analyses in the present study. We estimated the SNP-heritability of PSC to be 0.23. Among serologic biomarkers, an increased alkaline phosphatase (ALP) level and conditions such as a blocked bile duct had an estimated SNP-heritability of 0.25. We also examined the magnitude and direction of shared genetic

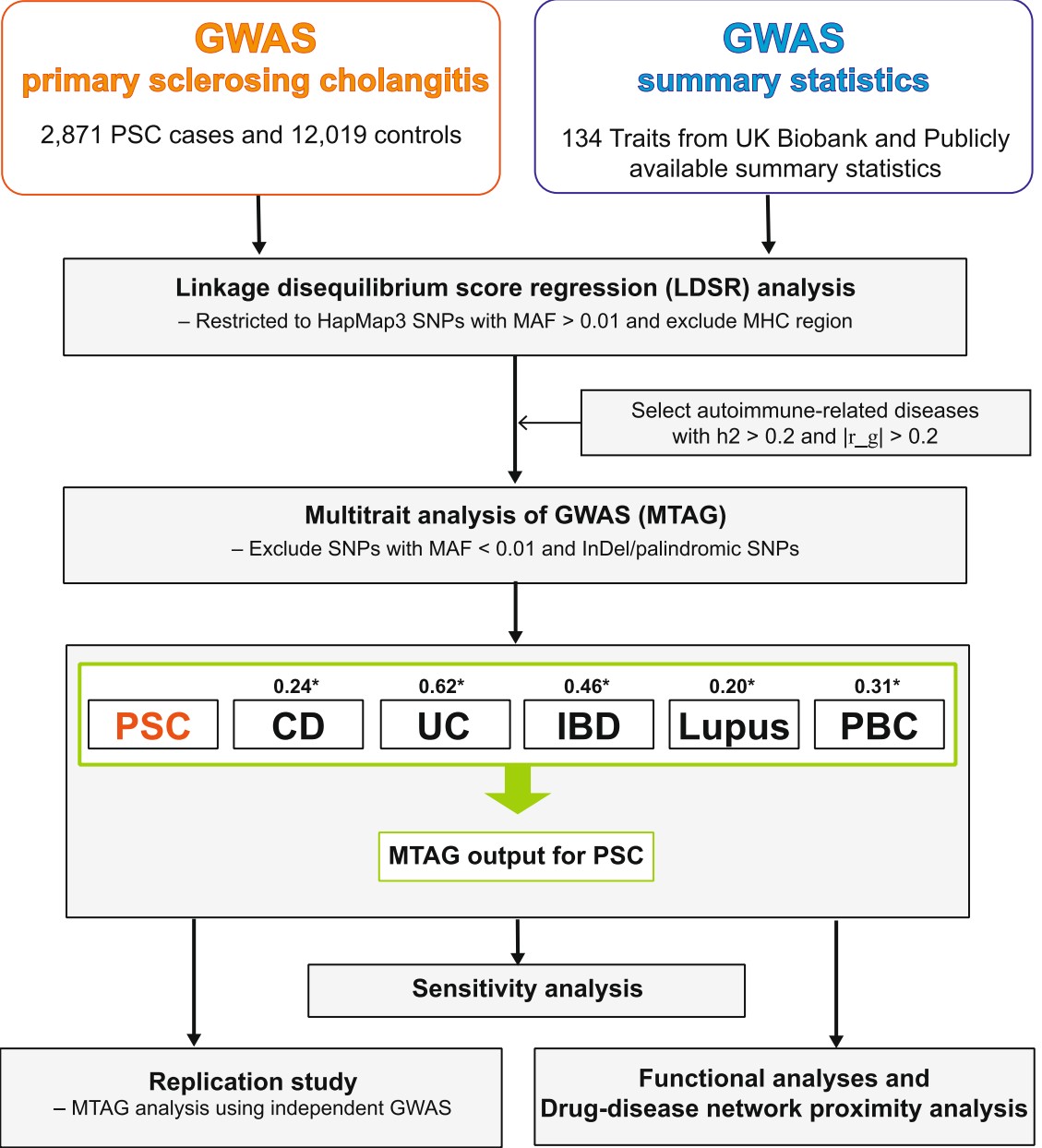

**Fig. 1 | Flow chart of the analytical workflow in this study.** h2 represents SNP-based heritability in the observed scale. |r_g| represents the absolute value of the pairwise genetic correlation between PSC and the traits studied. MAF stands for a minor allele frequency. MHC region stands for the major histocompatibility complex region. The asterisk "*" indicates the genetic correlation between PSC and each tested trait. The imputed summary statistics for PBC were used for subsequent analyses.

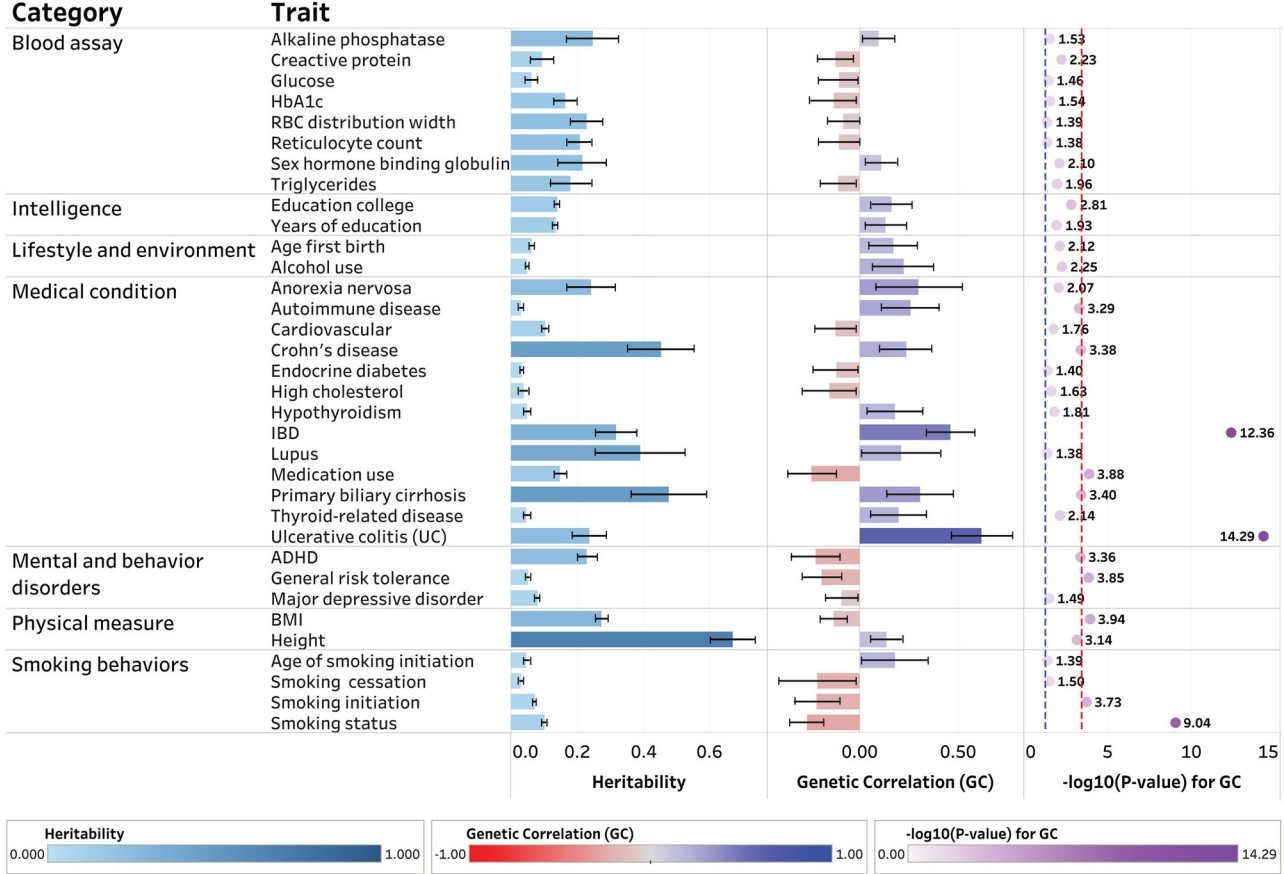

**Fig. 2 | The shared heritability and genetic correlation of PSC among clinical and epidemiological traits.** The dotted lines in blue and red indicate nominally and Bonferroni-corrected significant levels of $-\log_{10}(0.05) = 1.30$ and $-\log_{10}\left(3.73 \times 10^{-4}\right) = 3.43$, respectively. The error bar represents 95% confidence interval for the estimate of SNP-based heritability and pairwise genetic correlation of PSC in each trait, respectively. Sample sizes used to derive the estimates of SNP-based heritability and pairwise genetic correlation of PSC in each trait are shown in Supplementary Data 1. The dashboard for visualizing the results from LDSR was created using Tableau Desktop software (version 2022.2).

contribution between PSC and 134 polygenic traits of clinical and epidemiological parameters based on the cross-trait genetic correlation (r_g). We identified several polygenic traits showing moderate to strong genetic correlation with PSC at a Bonferroni-corrected significance level of $P = 0.05/134 = 3.73 \times 10^{-4}$. Since this is hypothesis-based research, we also considered $P < 0.05$ to identify nominally significant associations that could be examined in future studies. We considered $P$-values less than the Bonferroni-corrected significance level to be robustly associated in this study and the highlighted traits are displayed in Fig. 2. Our findings reported in Supplementary Data 3 demonstrated that the genetic architecture of PSC susceptibility was positively correlated with that of several immune-related diseases including IBD (r_g = 0.46; $P = 4.41 \times 10^{-13}$), UC (r_g = 0.62; $P = 5.18 \times 10^{-15}$), CD (r_g = 0.24; $P = 4.16 \times 10^{-4}$), lupus (r_g = 0.21; $P = 0.04$), and PBC (r_g = 0.31; $P = 3.95 \times 10^{-4}$). Overall shared genetic contribution between PSC and a behavior parameter, general risk tolerance defined as the willingness to take risks[22], showed a significant negative correlation (r_g = −0.20; $P = 1.41 \times 10^{-4}$). Increased body mass index (BMI) had a significant negative genetic correlation with PSC susceptibility (r_g = −0.13; $P = 1.16 \times 10^{-4}$). In epidemiological studies[7,23,24], the association between PSC and cigarette smoking has been inconsistent. Among traits related to smoking behaviors in this study, smoking status[25] modeled in previous smokers versus current smokers showed a strong negative genetic correlation with PSC susceptibility (r_g = −0.27; $P = 9.17 \times 10^{-10}$) while smoking initiation[26], which is a binary phenotype indicating whether an individual had ever smoked regularly (i.e., never-smokers versus ever-smokers), reported a

significant negative genetic correlation with PSC (r_g = −0.20; $P = 2.05 \times 10^{-6}$).

## MTAG with immune-mediated diseases identifies new PSC-associated loci with evidence of replication

Based on findings from the genome-wide SNP-heritability and pairwise genetic correlation, we restricted our MTAG to the traits for which LDSR has suggested strong associations with PSC susceptibility, showing h2 > 0.20 and |r_g| > 0.20 ("Methods", Supplementary Information). Five autoimmune-related disorders, CD (r_g = 0.24), UC (0.62), IBD (0.46), lupus (0.20), and PBC (0.31) were selected to identify new PSC risk loci using MTAG (Table 1). Compared to the conventional univariate GWAS, we detected more significant and stronger PSC-specific association signals when implementing MTAG. From MTAG combining PSC with five immune-related diseases; CD, UC, IBD, lupus, and PBC, we discovered seven loci (2p16.1, 4q24, 6q21.2, 6q23.3, 7q32.1, 10q24.2, and 16q22.1) that have not been previously reported or failed to reach the genome-wide significance level and one new independent significant variant of the reported locus (3p21.31) at the genome-wide significance level of $5.0 \times 10^{-8}$ (Table 2 and Fig. 3). In addition, our MTAG-identified PSC-specific results confirmed 11 PSC-specific risk-associated variants that have been previously reported in a single-disease GWAS of PSC susceptibility. These include genetic variants from well-established risk loci at 1p36.32, 2q33.2, and 6p21.33-p21.32 that are strongly associated with autoimmune-related diseases[2,20,27,28]. We displayed a Manhattan plot for the MTAG-identified PSC-specific GWAS (MTAG_PSC, Fig. 3b) along

**Table 1 | Estimate of genetic correlation among autoimmune-related diseases**

|                                          | PSC | CD               | UC          | IBD         | Lupus         | PBC*        |
|------------------------------------------|-----|------------------|-------------|-------------|---------------|-------------|
| Primary sclerosing cholangitis (PSC)     | 1   | 0.24 (se = 0.07) | 0.62 (0.08) | 0.46 (0.06) | 0.20 (0.10)   | 0.31 (0.09) |
| Crohn's disease (CD)                     |     | 1                | 0.62 (0.03) | 0.92 (0.02) | 0.13 (0.055)  | 0.18 (0.05) |
| Ulcerative colitis (UC)                  |     |                  | 1           | 0.90 (0.01) | 0.22 (0.07)   | 0.23 (0.05) |
| Inflammatory bowel disease (IBD)         |     |                  |             | 1           | 0.19 (0.05)   | 0.23 (0.04) |
| Lupus                                    |     |                  |             |             | 1             | 0.49 (0.06) |
| Primary biliary cirrhosis (PBC)*         |     |                  |             |             |               | 1           |

The asterisk "*" indicates that imputed summary statistics were used to estimate the SNP-heritability and pairwise genetic correlation using the SSimp package. "se" stands for the standard error of the pairwise genetic correlation between PSC and each trait.

with that from the previously published single-disease GWAS of PSC[2] (GWAS_PSC, Fig. 3a). There was no substantial evidence for inflation of both GWAS test statistics ($\lambda_{\text{GWAS\_PSC}} = 1.06$; $\lambda_{\text{MTAG\_PSC}} = 1.08$) shown in Fig. 3c, d, respectively. MTAG-identified genomic risk variants associated with PSC susceptibility with a $P < 5.0 \times 10^{-8}$ are reported in Supplementary Data 4.

A newly identified association of an intronic variant, rs228614, was detected in *MANBA* on 4q24 ($P_{\text{MTAG\_PSC}} = 1.71 \times 10^{-9}$). Associations at *MANBA* have been previously reported for multiple sclerosis[29], primary biliary cirrhosis[30], psoriasis[31], numerous hematologic traits[32–35], asthma[36,37], and major depressive disorders[38]. Another association at rs17780429 between *TNFAIP3* and *LINC02528* on 6q23.3 showed a strong genetic signal ($P_{\text{MTAG\_PSC}} = 2.24 \times 10^{-10}$) and many associations at *TNFAIP3* have been observed in autoimmune-related diseases[39–42] and multiple blood-cell traits[34,43]. We found a new intergenic variant, rs3757387 between *KCP* and *IRF5* on 7q32.1 ($P_{\text{MTAG\_PSC}} = 2.19 \times 10^{-14}$). rs3757387 has been previously reported for significant associations with systematic lupus erythematosus among diverse populations[44] and in a single population[19,45], rheumatoid arthritis in multiple populations[46,47], and Sjögren's syndrome[48]. An *NKX2-3* intronic variant, rs791168 on 10q24.2, was associated with PSC susceptibility and has been reported in many autoimmune-related and blood-cell traits[13] ($P_{\text{MTAG\_PSC}} = 1.33 \times 10^{-8}$). LocusZoom regional plots of genome-wide associations for these newly identified loci are provided in Supplementary Fig. 1.

To assess whether our MTAG results were robust to strong genetic correlation and clinical relevance among IBD, UC, and CD, we repeated our MTAG analysis only including PSC, CD, UC, lupus, and PBC (MTAG_PSC⊥IBD) as a sensitivity analysis. The results from the MTAG-identified PSC-specific model excluding IBD were very similar to those from the inclusion model (MTAG_PSC) (Table 2 and Supplementary Fig. 2).

To replicate the new MTAG-identified PSC-specific associations, we downloaded GWAS summary statistics from FinnGen[14] and GWAS Catalog[13], which are independent GWAS from the discovery phase (Supplementary Data 2). Since we were interested in replicating eight new associations (seven newly identified loci and one independent significant variant in the reported locus), we did not apply multiple testing corrections. We replicated four PSC-specific associations (MTAG_PSC_R), rs6787808 in *QRICH1* ($P_{\text{MTAG\_PSC\_R}} = 1.79 \times 10^{-2}$), rs228614 in *MANBA* ($P_{\text{MTAG\_PSC\_R}} = 2.05 \times 10^{-2}$), rs3757387 between *KCP* and *IRF5* ($P_{\text{MTAG\_PSC\_R}} = 1.39 \times 10^{-8}$), and rs791168 in NKX2-3 ($P_{\text{MTAG\_PSC\_R}} = 1.20 \times 10^{-3}$) at the nominal significance level of 0.05 (Table 2 and Supplementary Fig. 2).

### Fine-mapping and functional annotation nominates candidate variants within MTAG-identified loci

To pinpoint genomic risk loci and prioritize susceptibility variants underlying the MTAG-identified PSC-specific GWAS associations by functional annotation, positional, expression quantitative trait loci (eQTL), and chromatin interaction mappings, we exploited Functional Mapping and Annotation of GWAS (FUMA GWAS)[49] using LD structure

based on European ancestry of 1000 Genome Project phase 3 ("Methods"). We prioritized 406 unique genes from 20 PSC susceptibility loci reported in Supplementary Data 5 that functionally mapped and annotated using MTAG-identified GWAS, of which 109 genes were identified by position mapping of deleterious coding variants with the combined annotation-dependent depletion (CADD) score (posMapMaxCADD ≥ 12.37)[50] (Supplementary Data 6). Out of 406 prioritized genes, 48 genes (12%) were detected by eQTL associated with the expression of 14 immune cell types[51]. In the chromatin interaction mapping, 278 genes (69%) are mapped to the regions interacting with the promoter of the listed gene and of which 90 genes (32%) were found in the liver tissue in which the chromatin interaction is observed (Supplementary Data 6). Either chromatin interactions or eQTLs within PSC risk loci (Supplementary Data 5) were shown on chromosomes 2, 3, 4, 6, 7, 11, 16, 19, and 21, respectively (Supplementary Figs. 3). Then, 158 genes were mapped by both eQTLs and chromatin interactions including *IRF5* and *TNPO3* genes (in red in Supplementary Fig. 3e) on the 7q32.1. In addition, we explored immune-related genes among 406 PSC-specific susceptibility genes prioritized by position, eQTL, or chromatin interaction mapping using InnateDB[52] ("Methods"). We found five immune-related genes including *IRF5* and *SMO* (7q32.1) and *HAS3*, *SNTB2*, and *VPS4A* (16q22.1), within newly identified loci that have not been previously reported (Supplementary Data 7).

To functionally characterize the 329 independent significant variants within 20 genomic risk loci generated from FUMA, we performed an integrated variant functional annotation approach using the Functional Annotation of Variants Online Resource (FAVOR) platform[53–55] and the multidimensional annotation class integrative estimator[56,57] (MACIE). Out of 168 noncoding genes, we observed 14 more likely deleterious genes (CADD PHRED ≥ 12.37) and 8 and 6 genes on promoter and permissive enhancer sites, respectively. (Supplementary Data 8 and 9). Of the SNPs investigated with MACIE, we find 80 variants with a regulatory class prediction greater than 95%. That is, these variants are highly likely to tangibly affect the behavior of certain gene expressions, most often nearby genes. We find four variants with a conserved class prediction greater than 95%, and three of these variants also possess a regulatory prediction greater than 95%. That is, the four variants are highly likely to belong to the class of evolutionarily conserved variants that are found in many living beings. The full predictions for each SNP can be found in Supplementary Data 10.

To nominate the candidate causal variants from each locus for further functional analysis, we implemented fine-mapping of MTAG-identified loci using FINEMAP[58] and surveyed credible sets of plausible causal variants based on posterior inclusion probability (PIP). We then applied Conditional and Joint Analysis (COJO) using GCTA[59] to refine independent associations with prioritized risk loci. Based on the single-SNP PIP with each locus, we identified 32 variants falling into the 95% credible set across eight MTAG-identified GWAS loci (Supplementary Data 11). We found that eight MTAG-identified PSC risk loci explained at least two independent association signals; 2p16.1 locus harboring *PUS10*, with five independent variants, 3p21.31 (*QRICH1*) and 4q24 (*MANBA*) with five variants, 6p21.2 (*KCNK17*) with two variants, 6q23.3

**Table 2 | The MTAG-identified new associations of PSC**

| SNP Information (Ji et al. 2017) | | | | MTAG_PSC (Discovery) | | MTAG_PSC⊥IBD (Sensitivity) | | MTAG_PSC_R (Replication) | | GWAS_PSC (Ji et al. 2017) | |
|---|---|---|---|---|---|---|---|---|---|---|---|
| SNP*:A1/A2 | Chr:Position | Cytoband | Gene | OR | P | OR | P | OR | P | OR (EAF) | P |
| rs7608697:C/A | 2:61204641 | 2p16.1* | PUS10 | 1.07 | $3.11 \times 10^{-9}$ | 1.07 | $3.04 \times 10^{-9}$ | 1.01 | $9.24 \times 10^{-1}$ | 1.14 (0.39) | $1.04 \times 10^{-4}$ |
| rs6778808:C/T | 3:49079105 | 3p21.31# | QRICH1 | 1.08 | $1.20 \times 10^{-9}$ | 1.08 | $1.08 \times 10^{-9}$ | 1.01 | $1.79 \times 10^{-2}$ | 1.04 (0.04) | $8.47 \times 10^{-8}$ |
| rs228614:A/G | 4:103578637 | 4q24* | MANBA | 0.94 | $1.71 \times 10^{-9}$ | 0.94 | $1.85 \times 10^{-9}$ | 0.99 | $2.05 \times 10^{-2}$ | 0.87 (0.53) | $1.26 \times 10^{-6}$ |
| rs12198665:G/T | 6:39240796 | 6p21.2* | KCNK17 | 0.94 | $3.35 \times 10^{-8}$ | 0.94 | $3.09 \times 10^{-8}$ | 1.00 | $9.18 \times 10^{-1}$ | 0.84 (0.30) | $7.99 \times 10^{-8}$ |
| rs1780429:A/G | 6:138222588 | 6q23.3* | TNFAIP3 | 0.91 | $2.24 \times 10^{-10}$ | 0.91 | $1.28 \times 10^{-10}$ | 1.00 | $1.23 \times 10^{-1}$ | 0.77 (0.85) | $1.16 \times 10^{-7}$ |
| rs3757387:C/T | 7:128576086 | 7q32.1* | IRF5 | 1.08 | $2.19 \times 10^{-14}$ | 1.08 | $1.54 \times 10^{-14}$ | 1.01 | $1.39 \times 10^{-8}$ | 1.13 (0.46) | $3.01 \times 10^{-4}$ |
| rs791680:C/A | 10:101293468 | 10q24.2* | NKX2-3 | 0.94 | $1.33 \times 10^{-8}$ | 0.94 | $1.23 \times 10^{-8}$ | 0.99 | $1.20 \times 10^{-3}$ | 0.88 (0.49) | $1.28 \times 10^{-5}$ |
| rs7939027:C/A | 16:68942590 | 16q22.1* | TANGO6 | 1.15 | $1.69 \times 10^{-8}$ | 1.15 | $1.39 \times 10^{-8}$ | 1.00 | $7.53 \times 10^{-1}$ | 1.35 (0.05) | $1.86 \times 10^{-6}$ |

Gene, the nearest genes ±200 kb of the genomic risk SNP (reference NCBI build37); A1/A2, effect allele/other allele; EAF, effect allele frequency; OR odds ratio; P, P-value; MTAG_PSC, MTAG-identified PSC-specific association modeled in the discovery phase (6 GWAS in total); MTAG_PSC⊥IBD, MTAG-identified PSC-specific association modeled in the sensitivity analysis (5 GWAS in total); MTAG_PSC_R, MTAG-identified PSC-specific association modeled in replicate phase (6 GWAS in total); GWAS_PSC, single-disease PSC GWAS; ORs are calculated from a joint meta-analysis using MTAG; Two-sided raw P-values are reported; *, the new risk loci identified in this study; #, the new lead variant from a previously reported locus with $r^2 < 0.1$.

(*TNFAIP3*) and 7q32.1 (*IRF5*) with five variants, 10q24.2 (*NKX2-3*) with three variants and 16q22.1 (*TANGO6*) with two variants, respectively. There is no additional genome-wide significant association from GCTA-COJO analysis at the genome-wide significant level of $5 \times 10^{-8}$.

**eQTL-based colocalization prioritizes PSC susceptibility genes from the MTAG-identified new loci**

We carried out eQTL-based colocalization analysis to identify allelic-specific effects on gene expression and to examine colocalization of association signals from new MTAG-identified PSC risk-associated findings using eQTL summary statistics of 49 tissue types from GTEx v8. Among seven MTAG-identified new risk loci (2p16.1, 4q24, 6p21.2, 6q23.3, 7q32.1, 10q24.2, 16q22.1), colocalization nominated three candidate genes, *MANBA* at 4q24, *IRF5* at 7q32.1, and *NKX2-3* at 10q24.2, contributing to PSC risk (Supplementary Data 12). Notably, a newly MTAG-identified locus, *IRF5*, displayed the highest posterior probability scores indicating that both PSC and each of the 30 tissues are associated and share a single functional variant (PP4 > 0.80) using coloc[60] package (Fig. 4, Supplementary Fig. 4, Supplementary Data 12).

We selected 406 prioritized genes to detect relevant groups of related genes involved in the regulation of specific biological pathways. Using STRING Protein–Protein Interaction (PPI) networks[61], these candidate genes are highly enriched for protein–protein interactions ($P < 1.00 \times 10^{-16}$), with enrichment at false discovery rate (FDR) < 0.05 of the following pathways: immune receptor activity (FDR = $3.84 \times 10^{-2}$), beta-2-microglobulin binding ($1.10 \times 10^{-2}$), cytokine-mediated signaling pathway ($1.58 \times 10^{-13}$), interferon-gamma-mediated signaling pathway ($1.13 \times 10^{-11}$), T-cell receptor signaling pathway($2.21 \times 10^{-11}$), immune response-activating cell surface receptor signaling pathway ($2.65 \times 10^{-9}$), interleukin-7-mediated signaling pathway ($9.21 \times 10^{-9}$), TNFR2 noncanonical NF-kB pathway ($7.90 \times 10^{-3}$), Th17 cell differentiation ($2.63 \times 10^{-6}$), and Th1 and Th2 cell differentiation ($1.94 \times 10^{-5}$) (Supplementary Data 13, Supplementary Fig. 5). For comparison, we implemented enrichment analysis using the Database for Annotation, Visualization, and Integrated Discovery (DAVID) Bioinformatics Resources[62,63] on the same candidate 406 genes. We observed T-cell receptor signaling pathway (FDR = $5.82 \times 10^{-7}$), antigen processing and presentation ($8.18 \times 10^{-15}$), immunoglobulin production involved in immunoglobulin mediated immune response ($6.30 \times 10^{-14}$), cytokine Signaling in Immune system ($3.48 \times 10^{-5}$), interferon Signaling ($2.62 \times 10^{-9}$), and interferon alpha/beta signaling ($6.60 \times 10^{-4}$) (Supplementary Data 14).

In addition, we scrutinized the PPI network associated with each gene prioritized from newly MTAG-identified loci and found three genes (*MANBA, IRF5*, and *NKX2-3*) to be highly enriched for PPI at FDR < 0.05. Each prioritized gene of *MANBA, IRF5*, and *NKX2-3* reported a PPI P-value of $5.16 \times 10^{-14}$, $1.00 \times 10^{-16}$, and $1.13 \times 10^{-9}$, respectively. We observed B and T-cell receptors, chemokine, C-type lectin receptor, cytosolic DNA-sensing, HIF-1, IL-17, JAK-STAT, MAPK, metabolic, NF-kappa B, NOD-like receptor, PD-L1 expression and PD-1 checkpoint in cancer, RIG-I-like receptor, th1-th2 cell differentiation, th17 cell differentiation, thyroid hormone, TNF, and toll-like receptor signaling pathways in the KEGG pathways at FDR < 0.05 using STRING PPI networks (Supplementary Data 15, Supplementary Fig. 6).

**Network-based proximity predicts drug-PSC associations for drug repurposing**

Although there is no medication proven to treat PSC, ursodeoxycholic acid (UDCA) is a recommended treatment increasing the bile flow as well as preventing damage to liver cells. While UDCA is used to treat PBC and radiolucent gallstones with a functioning gall bladder, it does not appear to improve survival or reduce the need for liver transplant in PSC patients. From in silico network-based proximity analysis[64], we estimated the shortest distance ($d$) between drug targets and PSC

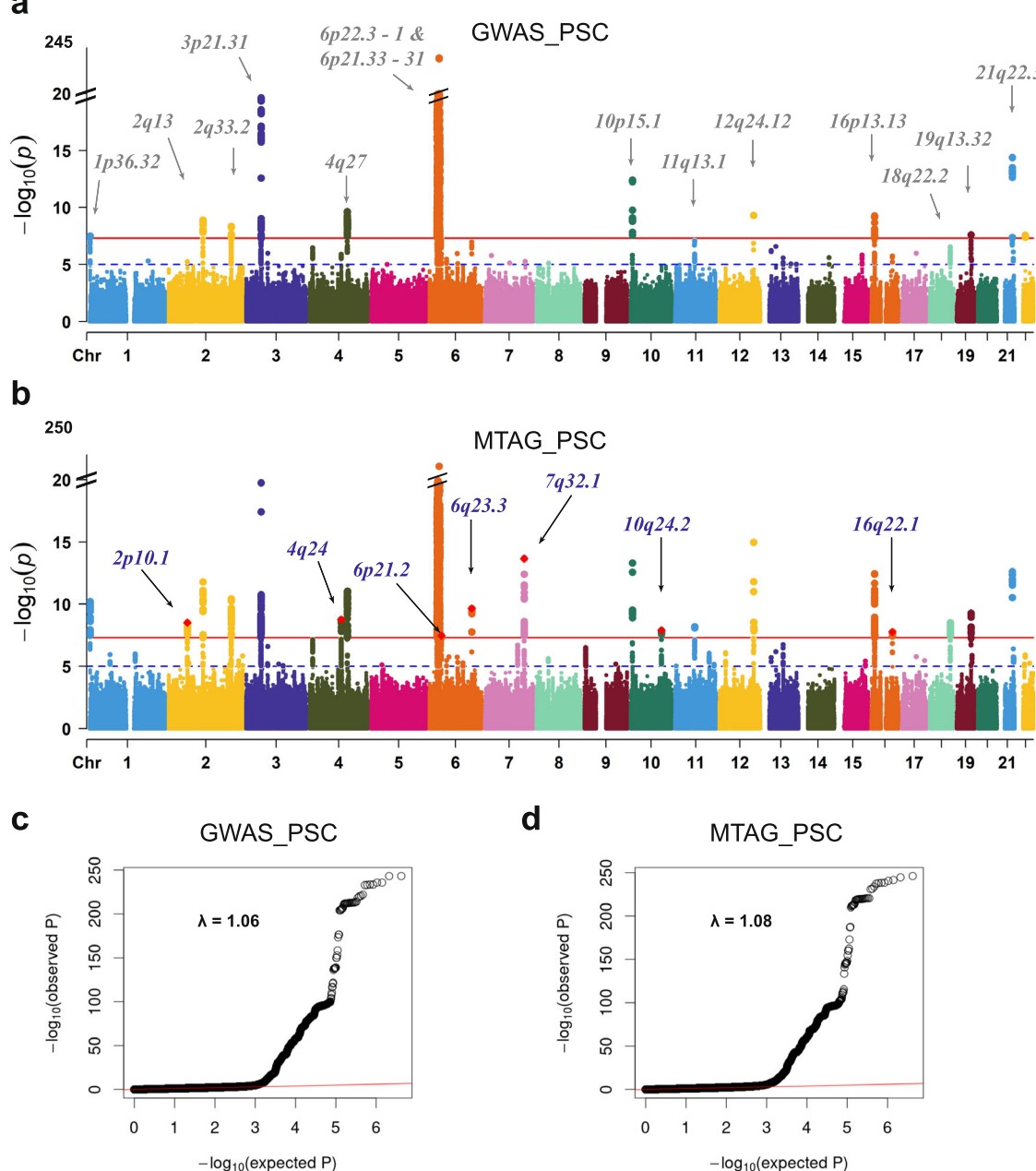

**Fig. 3 | Manhattan plots and quantile-quantile plots for the single-trait GWAS and the multitrait GWAS of PSC. a, c** PSC single-trait GWAS (Ji et al., 2017, PMID:27992413; GWAS_PSC). **b, d** MTAG-identified PSC-specific GWAS against five immune-mediated disorders, CD, UC, IBD, lupus, and PBC (MTAG_PSC). The *x*-axis represents chromosomal location, and the *y*-axis represents the −log10(*P*-value). The cytoband annotations for the newly and previously identified loci are in purple (**b**) and gray (**a**), respectively. The solid lines in red and the dotted lines in blue indicate the genome-wide significant two-sided unadjusted *P*-value of −log10($5 \times 10^{-8}$) and the suggestive significant two-sided unadjusted *P*-value of −log10($1 \times 10^{-5}$), respectively. *P*-values are derived using multitrait analysis of GWAS in the discovery study.

candidate genes (Supplementary Data 16, "Methods") and the relative proximity measure(*z*) capturing the statistical significance of distance between drug and disease protein derived from a permutation test (Table 3, Supplementary Data 17, Supplementary information). The more negative the relative proximity between drug and disease, the closer the genetic relationship between them[64]. We identified many agents at the relative proximity threshold of −0.15, implying potential therapeutic effects on PSC. The top-ranked drugs suggestive for PSC included denileukin diftitox, interleukin-2-alpha binder used for cutaneous T-cell lymphoma (*z* = −5.443); vitamin E (*z* = −1.918); MLN0415, a

small molecule IKK2 inhibitor downregulating the expression of a number of inflammatory proteins (*z* = −1.648). The proximity of UDCA showed 0.170 on PSC indicating that it may not be a genetically promising candidate drug for PSC. The FUMA platform facilitates gene mapping to the DrugBank database via GENE2FUNC reported in Supplementary Data 18. While network-based proximity predicts drug association based on the distance between drug targets and candidate genes, FUMA provides the gene table mapped to the drug database based on the prioritized genes by different mapping methods such as position, eQTL, and chromatin interaction.

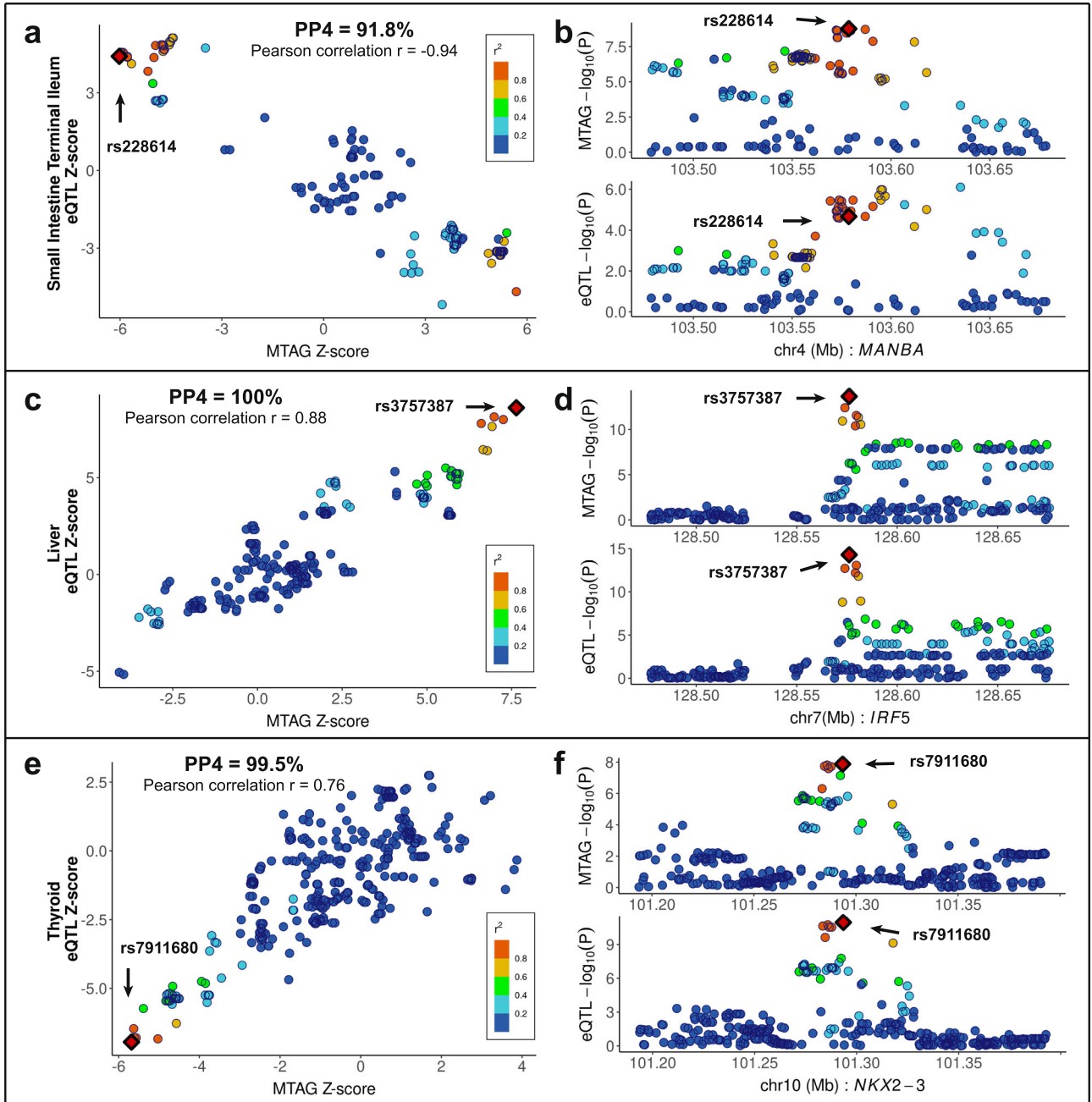

**Fig. 4 | Functional validation of the MTAG-identified PSC-specific candidate genes. a, c, e** eQTL signals in GTEx v8 small intestine terminal ileum (*n* = 174) for *MANBA* (**a**), liver (*n* = 208) for *IRF5* (**c**), and thyroid (*n* = 574) for *NKX2-3* (**e**) colocalize with those of the MTAG-identified PSC-specific GWAS by coloc (posterior probability for the same causal variant shared between MTAG-identified GWAS and a tissue-specific eQTL (PP4) = 0.918 for rs228614, PP4 = 1.00 for rs3757387, and PP4 = 0.995 for rs7911680), respectively. Pearson correlation (*r*) is shown between the *Z*-score of eQTL (*y*-axis) and MTAG_PSC (*x*-axis). Variants are color-coded based on the LD r2 (1000 Genomes phase 3, EUR) with the candidate variants (red dot in a diamond shape). Variants with imputation quality scores >0.6 were plotted in this region. **b, d, f** Regional association plots of eQTL and MTAG_PSC within ±100kb of rs228614 (**b**), rs3757387 (**d**), and rs7911680 (**f**) are displayed.

## Discussion

We leveraged publicly available GWAS summary statistics to investigate the shared genetic architecture of PSC with a variety of clinical and epidemiological traits and to identify additional PSC-risk loci. We first scrutinized the patterns of genomic overlap between PSC and numerous phenotypes using LDSR. Cross-trait LDSR estimated the genetic correlation between traits to gain insights into common etiologies[15,16]. We identified significant phenotypic associations between different polygenic traits and PSC. The findings of this study enabled us to confirm previously well-established comorbid conditions and to identify polygenic traits for further study. Complementary

approaches such as MTAG, which is a joint association analysis of genetically correlated traits, helped us to discover new susceptibility variants influencing PSC. In addition, LDSR-identified polygenic traits indicating a high correlation with PSC can be applied in Mendelian randomization analysis to unveil further causal relationships between PSC and the traits of interest.

We observed a significant positive correlation between the genomic architecture of each autoimmune-related disease and that of PSC using LDSR. In several genetic studies, PSC is driven by shared and distinct genetic determinants compared to immune-mediated diseases[7,19,27,65,66]. The shared structure of the genetic susceptibility to

**Table 3 | Network-based in silico drug repurposing on PSC**

| DrugBank id | Drug Name | Description | Indication | z | p |
|---|---|---|---|---|---|
| DB00004 | Denileukin diftitox | CD25-directed cytotoxin | Cutaneous T-cell lymphoma | −5.443 | $2.63 \times 10^{-8}$ |
| DB05299 | Keyhole limpet hemocyanin* | Immune modulator | Bladder cancer, solid tumors | −4.020 | $2.91 \times 10^{-5}$ |
| DB06584 | TG4010* | Cancer vaccine expressing MUC1/IL2 | Breast cancer, renal cell carcinoma, prostate cancer, non-small cell lung cancer. | −3.561 | $1.85 \times 10^{-4}$ |
| DB05304 | Girentuximab* | Chimeric monoclonal antibody targeting carbonic anhydrase IX | Renal cell carcinoma | −3.526 | $2.11 \times 10^{-4}$ |
| DB06083 | Tapinarof* | Aryl hydrocarbon receptor-modulating agent | Plaque psoriasis | −3.013 | $1.29 \times 10^{-3}$ |
| DB04901 | Galiximab* | Anti-CD80 monoclonal antibody | Non-Hodgkin's lymphoma, psoriasis | −2.740 | $3.07 \times 10^{-3}$ |
| DB00163 | Vitamin E | Vitamin | Dietary supplement | −1.918 | 0.0276 |
| DB06421 | Declopramide* | DNA repair inhibitors | Colorectal cancer, inflammatory bowel disease. | −1.593 | 0.0556 |
| DB06362 | Becatecarin* | DNA intercalating agent, topoisomerase I and II inhibitor | Gastric cancer, adenocarcinoma of unknown origin, gall bladder or pancreatic tumors, breast cancer, renal cell cancer, colorectal cancer | −1.542 | 0.0615 |
| DB05022 | Amonafide* | DNA intercalating agent, topoisomerase II inhibitor | Breast cancer, ovarian cancer, prostate cancer, acute myeloid leukemia | −1.542 | 0.0616 |
| DB08934 | Sofosbuvir | N55B RNA polymerase inhibitor | Chronic hepatitis C infection | −1.342 | 0.0898 |
| DB05127 | ANA971* | Toll-like receptor 7 | Chronic hepatitis C infection | −1.338 | 0.0904 |
| DB04860 | Isatoribine* | Toll-like receptor 7 | Chronic hepatitis C infection | −1.338 | 0.0904 |
| DB11094 | Vitamin D | Vitamin | Osteoporosis prevention, Vitamin D insufficiency/deficiency, hypoparathyroidism, refractory rickets, familial hypophosphatemia | −0.515 | 0.3033 |
| DB01586 | Ursodeoxycholic acid | Gallstone dissolution agent | Gallstones, PBC | 0.170 | 0.5673 |

DrugBank id, DrugBank database identifier; Drug Name, drug name; Indication, current drug-treatment; z, relative proximity between PSC candidate genes and relevant drug of genes; p, P-value of the relative proximity. *Asterisk indicates not FDA-approved agent.

PSC is notably overlapped with immune-mediated disorders such as CD, IBD, lupus, PBC, and UC[27], which have well-established associations with PSC[67]. In addition, these immune-mediated disorders showed large proportions of phenotypic variance explained by all common SNPs in this study.

Several epidemiological studies have reported inverse associations between smoking and PSC risk[7,23,24,68,69]. Our study found a strongly protective genetic correlation between the genomic architecture of smoking status modeled in former smokers versus current smokers and that of PSC, suggesting that the genetic contribution of current smoking is associated with a decreased risk of PSC compared to that of former smoking. Although it failed to meet the Bonferroni-corrected significance level of $3.73 \times 10^{-4}$, the smoking cessation trait modeled in former smokers versus current smokers[26] showed a consistent association with PSC implying that the genetic contribution of current smoking is associated with a decreased risk of PSC compared to that of former smoking[23]. The smoking initiation trait modeled in never-smokers versus ever-smokers[26] showed a significant negative association with PSC suggesting that PSC risk among current and former smokers is significantly lower than that among never-smokers[23]. Smoking promotes chronic epithelial and tissue injury through chronic airway inflammation[70,71] and the most common causes of chronic inflammation include immune-mediated disorders which could potentially contribute to PSC development. Therefore, the shared association of PSC with smoking behaviors makes disentangling such effects challenging.

Applying an orthogonal genomics-driven method complementing clinical epidemiologic research of PSC, we confirmed a link between PSC risk and elevated BMI and diabetes[7,72–75]. However, clinical studies have shown inconsistent associations between cardiovascular disease and PSC[75,76]. Pairwise genetic correlation between PSC and cardiovascular risk demonstrated a negative association at the nominal significance level of 0.05. We also identified several suggestive polygenic traits for which the pairwise genetic correlations were nominally significant at $P < 0.05$. We observed a nominally significant

inverse genetic correlation between PSC and several serologic biomarkers including C-reactive protein, glucose, HbA1c, red blood cell distribution width, reticulocyte count, and triglycerides while alkaline phosphatase and sex hormone binding globulin were positively correlated with PSC risk. These findings through LDSR show good concordance with previous clinical and genetic epidemiologic studies[7,75,77].

Implementing MTAG, we discovered seven new susceptibility loci that have not been previously reported in GWAS_PSC and, of these, we replicated three lead associations in other GWAS independent from the discovery phase. Two of the new MTAG PSC loci, *MANBA* on 4q24 and *IRF5* on 7q32.1 were previously shown to be associated with several hematology-related traits and immune-mediated disorders[20,44–48]. The previously identified phenotypes have also been reported in PSC. In addition, we prioritized candidate genes for PSC susceptibility through MTAG and inferred biological pathways identified through eQTL-colocalization analyses. PPI networks showed that candidate genes were often part of biological pathways involving metabolic processes and immune response.

Recently, the identification of targets for drug repurposing (repositioning) using genome-wide approaches has become popular[20]. In this study, we implemented network-based in silico drug efficacy screening to predict agents potentially suitable for repurposing to PSC. Generally, UDCA is recommended for the treatment of cholestatic liver diseases including PSC, but it does not show any effect on the progression and survival of PSC patients[78]. Interestingly, the proximity of UDCA shows that it may not be a genetically promising candidate drug for PSC. In clinical trials in the U.S., UDCA did not improve the management of PSC[79] and its use has been discouraged in the U.S. providers[80], indicating a correct prediction of our drug screening analysis. The identified candidate drugs are relevant to lymphoma (Denileukin diftitox, Galiximab), various cancers (Keyhole limpet hemocyanin, TG4010, Girentuximab, Amonafide), psoriasis and psoriatic disorders (Tapinarof), vitamin E, IBD (Declopramide), metabolic disorders (Girentuximab), rheumatoid arthritis, liver cancer (Becatecarin), chronic hepatitis C virus (HCV) (Sofosbuvir, ANA971,

Isatoribine). Poch et al. reported a single-cell atlas of intrahepatic T-cell landscape in PSC[81]. The top-ranked drug, Denileukin diftitox, which is involved in the regulation of immune tolerance by controlling regulatory T-cells activity, could be a candidate agent for further study of pharmacological effect.

Integration, harmonization, and optimization of the existing large-scale GWAS datasets have become a popular analytical strategy to identify new genetic associations. However, access to individual-level GWAS datasets remains limited due to data use restrictions. Although LDSR can quantify the shared genetic architecture of traits having undergone GWAS analysis without requiring GWAS individual-level data, it assumes an absence of population stratification in the underlying summary statistics of the tested traits and necessitates the incorporation of GWAS data from populations expected to have homogeneous genetic structure. Furthermore, GWAS summary statistics with small sample sizes or low SNP-heritability are not amenable to LDSR. One caveat of implementing LDSR is that nonsignificant associations could be due to limited statistical power, rather than a lack of shared heritability, as cross-trait LDSR requires larger sample sizes of GWAS summary-level data to achieve equivalent standard error compared to methods that use individual-level data[15]. Another limitation of LDSR is that the analysis includes only common genetic variants with MAF >0.01 and therefore fails to capture shared heritability due to underlying rare variants between PSC and multiple polygenic traits.

MTAG[21] can substantially improve statistical power for detecting susceptibility loci relative to separate GWAS for the traits tested and allows potential sample overlap in numerous trait-specific summary statistics from large-scale cohort GWAS. However, replication or validation analysis is recommended to assess the credibility of each SNP association when MTAG is applied to low-powered GWAS or to GWAS that are considerably heterogeneous in statistical power. Since MTAG uses overlapping SNPs across all GWAS summary statistics, combining summary statistics with a smaller number of SNPs with those with a larger number of SNPs can reduce statistical power.

In conclusion, our findings from LDSR confirm the associations between immune-mediated disorders and PSC, and epidemiological parameters associated with PSC susceptibility. We also identified and replicated the newly MTAG-identified PSC risk loci and through eQTL-colocalization analysis helped to prioritize candidate genes for PSC susceptibility. This study emphasizes the strong evidence that exists for the shared genetic underpinning among immune-mediated diseases. While PSC GWAS have identified a few risk-associated variants, the function and identity of the causal variants are not fully explored. To address the impact of PSC risk-associated variants in the immune system and within less-well-established noncoding regions, we highlighted several in silico functional approaches to map and prioritize the variants identified. Furthermore, we exploited an immune-related gene database for deciphering how PSC risk-associated variants may alter immune networks. We also utilized the integrative functional annotations platform to functionally characterize the prioritized genes including both coding and noncoding genes, which provide numerous information on variant and indel functional annotations. Since there is no medication proven to treat PSC, we predicted many potential agents at the relative proximity capturing the statistically significant relationship between a potential drug and putative disease-associated proteins. We further carried out gene mapping to the drug database with the broad range of genes prioritized by position, eQTL, and chromatin interaction mapping. These analytical pipelines, which utilize activity maps of noncoding regions help us pinpoint their role in specific cell types. These findings can provide better functional insight into the genetic etiology of PSC susceptibility and improve our understanding of how PSC risk-associated variants alter the immune system. Finally, future studies using causal inference approaches such as Mendelian randomization or genetic instrumental variable methods may help to elucidate the causal relationship between the risk of PSC and other potential candidate phenotypes to reveal surrogate biomarkers that may improve the predictive power of polygenic risk scores.

## Methods

### Ethics statement
All participants for each GWAS were recruited following protocols approved by the local Ethics Committee/Institutional Review Boards. Written informed consent was obtained from each participant included in the study. All methods were performed in accordance with the ethical guidelines of the 1975 Declaration of Helsinki.

### GWAS summary statistics and imputation
We obtained the GWAS summary statistics for PSC[2] and 134 clinical and epidemiological traits from existing data resources[12,13]. More details are shown in Supplementary Data 1 and Supplementary Information. We restricted the study populations to individuals of European ancestry to align with the homogeneous ancestry background of participants in GWAS of the traits tested in our downstream analyses. To enhance adequate statistical power in this study, GWAS summary statistics were imputed using the SSimp software[82] (v.0.5.6; https://github.com/zkutalik/ssimp_software) when the number of SNPs in a trait was considerably smaller compared to that in other traits, thus becoming less informative. Detailed methods are provided in Supplementary Information.

### Analyses of multitrait GWAS
We estimated SNP-heritability (h2) on the observed scale and pairwise genetic correlation (r_g) between multiple polygenic traits using LDSR[8–11,15,16] (v1.0.1; https://github.com/bulik/ldsc). We conservatively set the test-wise significance level using Bonferroni correction to be 0.05/134, adjusting for the analysis of 134 polygenic traits in total (Supplementary Information).

The commonly used conventional GWAS approach is to analyze the univariate association test for a single trait/phenotype. This does not permit leveraging of genetic information from other polygenic traits. Integrating associations from other traits highly correlated with PSC can improve the statistical power to identify new polygenic variants[21,83–85]. We conducted MTAG (v1.0.8; https://github.com/JonJala/mtag) combining PSC with immune-mediated disorders selected by h2 > 0.20 and |r_g| > 0.20. MTAG was modeled for PSC versus five polygenic autoimmune-related traits: CD, UC, IBD, lupus, and PBC (MTAG_PSC). Additionally, we performed a sensitivity analysis excluding IBD (⊥IBD) from the MTAG analysis (MTAG_PSC⊥IBD) since IBD is the umbrella term mainly comprising of medical conditions under which both CD and UC fall[86]. The sensitivity analysis included only five autoimmune-related diseases; PSC, CD, UC, lupus, and PBC.

To replicate MTAG-identified PSC risk-associated new loci, we implemented MTAG (MTAG_PSC_R) using PSC (FinnGen phenocode:K11_CHOLANGI), CD (K11_CD_NOUC), UC (K11_UC_NOCD), IBD (K11_IBD), and lupus (M13_SLE) from FinnGen repository[14], and PBC[87] from GWAS catalog, which are independent of those in the discovery phase. Details are reported in Supplementary Data 2.

### Characterization of genomic risk loci using FUMA
We mapped the genomic regions of associations by the most significant variants using FUMA GWAS[49] (v1.4.1; https://fuma.ctglab.nl/) platform computing LD structure, annotating functions to SNPs, and prioritizing candidate genes from MTAG-derived summary statistics[49]. To define genomic risk loci for MTAG-identified PSC susceptibility, we used linkage disequilibrium structure based on the European ancestry of the 1000 Genome Project phase 3. Genomic risk loci and the subsets of significant SNPs within each locus were identified using the SNP2GENE function applying the default thresholds: (1) independent

significant SNPs, defined as $P < 5 \times 10^{-8}$ and independent from each other at $r^2 \geq 0.6$ (2) lead SNPs, defined as independently significant SNPs and independent from each other at $r^2 \geq 0.1$; (3) genomic risk loci, defined by merging lead SNPs within physically overlapped LD blocks and all SNPs in linkage disequilibrium of $r^2 \geq 0.6$ with one of the independent SNPs. Prioritized susceptibility variants from MTAG GWAS were mapped by positional, eQTL, and chromatin interaction mappings using the FUMA SNP2GENE function with default settings. Finally, FUMA maps the prioritized genes given by the SNP2GENE function to the drug database (DrugBank[88]) via the GENE2FUNC function in the FUMA platform. The gene table mapped to the DrugBank database provides gene information and the relevant DrugBank IDs that can be found at https://go.drugbank.com/drugs with the details.

### Functional annotation within immune-related genes using InnateDB Innate Immunity Genes
We examined 406 prioritized genes to nominate innate immune genes associated with PSC using 7476 genes involved in innate immune responses from the InnateDB[52] portal. InnateDB provides the manually-curated list of genes and signaling responses involved in human innate immunity from publicly available databases including the Immunology Database and Analysis Portal (ImmPort) system, Immunogenetic Related Information Source (IRIS), MAPK/NFKB Network, and Immunome Database. The details can be found elsewhere at https://www.innatedb.com/redirect.do?go=resourcesGeneLists.

### Integrative multi-omic annotation analysis
We annotated the 406 prioritized genes using FAVOR platform[53–55] (v2.0; https://favor.genohub.org/) which is an open-access variant functional annotation portal for whole WGS/WES data. FAVOR provides functional annotation information of 8,812,917,339 SNVs across the human genome and 79,997898 indels from the Trans-Omics for Precision Medicine (TOPMed) BROVO variant set (Build GRCh38) based on a collection of databases such as variant category, evidence of chromatin, protein function, conservation, and Clinvar information. The details have been described elsewhere[55].

### Annotation-informed function prediction
We utilized the multidimensional annotation class integrative estimator[56,57] (MACIE, https://github.com/ryanrsun/lungCancerMACIE/tree/master/MACIE_pipeline) to analyze functional annotation data and understand the possible mechanistic roles of individual SNPs. For each variant, MACIE utilizes a generalized linear mixed model that specifies annotation values as outcomes and unobserved latent functional classes as predictors. The posterior probabilities of these unobserved classes are then calculated for each SNP to estimate the probabilities of possessing certain functions. The calculation proceeds through an expectation-maximization (EM) algorithm until convergence. The final posterior expected value of a class is taken as the MACIE prediction. Specifically, we applied MACIE with two latent classes, (1) regulatory class informed by 28 annotations such as H3K27Ac levels and (2) conserved class informed by eight phylogenetic conservational algorithms. Predictions were only made for noncoding variants.

### Fine-mapping and gene-based enrichment analyses
We implemented FINEMAP[58] (v1.4.1; http://www.christianbenner.com) to survey credible sets of plausible causal variants based on the posterior inclusion probability (PIP). We carried out the FINEMAP package with the options "--sss" to specify the "fine-mapping with shotgun stochastic search" and "--n-causal-snps 5" to set the maximum number of causal variants allowed within a locus to 5. We performed Conditional and Joint analysis using GCTA[59] (v1.9.4; https://cnsgenomics.com/software/gcta/) to select independent association signals within the prioritized risk loci with the option "--cojo-cond".

The Genotype-Tissue Expression (GTEx_v8)[89] database consists of data from 49 normal tissues from 838 donors (Supplementary Data 5, Supplementary information). Colocalization between the seven MTAG_PSC associations within the newly identified loci and eQTL signals was calculated using the coloc package (v5.1.0; https://cran.r-project.org/web/packages/coloc/)[60]. We focused on the colocalizations when coloc suggested a plausible posterior probability that both PSC and a tissue from GTEx_v8 are associated and share a single functional variant (PP4 > 0.80).

We utilized the STRING Database[61] (v11.5; https://string-db.org/cgi/input?sessionId=bmwWOuutn8ZR) to explore the functional enrichment of protein–protein interaction (PPI) networks and to scrutinize the enrichment of various pathways among the prioritized genes (proteins). In addition, we surveyed the DAVID Bioinformatics Resources[62,90] (v6.8; https://david.ncifcrf.gov/) to look for enrichment of various functional annotations on the 416 prioritized genes after excluding 9 overlapped genes from 19 newly MTAG-identified and previously reported PSC risk-associated genes and 406 genes mapped from position mapping, eQTL mapping, and chromatin interaction mapping provided from FUMA.

### Network-based proximity between drugs and disease-identified proteins for drug repurposing
Drug–disease proximity measures, distance ($d$), and the corresponding relative proximity ($z$), quantifying the network-based relationship between drugs and proteins encoded by genes associated with the disease while correcting for the known biases of the interactome[64], were estimated (Supplementary Information). To elucidate the effectiveness of proximity as an unbiased measure of drug–disease relatedness, we defined a drug to be proximal to a disease when the closest proximity, $z \leq -0.15$, and not proximal otherwise[64]. We downloaded detailed drug data with comprehensive drug target information from the DrugBank database (v5.1.9, released 2022-01-04)[88].

### Reporting summary
Further information on research design is available in the Nature Portfolio Reporting Summary linked to this article.

## Data availability
The summary statistics of PSC from MTAG are publicly available at https://github.com/biomedicaldatascience/PSC_MTAG. The GWAS summary-level data analyzed in this study are available in the NHGRI-EBI GWAS Catalog [https://www.ebi.ac.uk/gwas/] and the MRC IEU OpenGWAS database [https://gwas.mrcieu.ac.uk/] for previously published GWAS summary statistics, Neale's lab repository for UK Biobank GWAS summary statistics [https://github.com/Nealelab/UK_Biobank_GWAS], and FinnGen repository for Finnish Biobank GWAS summary statistics r6 [https://finngen.gitbook.io/documentation/v/r6/data-download]. The accessible links and reference information for the GWAS summary-level data (mapped to Genome Assembly GRCh37) used in this study can be found in Supplementary Data 1 and 2. Non-commercial DrugBank datasets (v5.1.9) are available and access can be obtained by the academic license [https://go.drugbank.com/releases/latest]. The data including all variant-gene cis-eQTL associations tested in each tissue (GTEx v8) are available in a requester pays bucket on Google Cloud Platform (GCP) [https://gtexportal.org/home/datasets; https://console.cloud.google.com/storage/browser/gtex-resources]. The immune-related genes can be obtained in the InnateDB portal [https://www.innatedb.com/redirect.do?go=resourcesGeneLists].

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

## Acknowledgements

We thank all individuals who have contributed their samples and clinical data for the PSC study, and we also thank the international PSC study group for sharing GWAS summary statistics of PSC. We want to acknowledge the participants and investigators of the FinnGen study. The Genotype-Tissue Expression (GTEx) Project was supported by the Common Fund of the Office of the Director of the National Institutes of Health, and by NCI, NHGRI, NHLBI, NIDA, NIMH, and NINDS. The data used for the analyses described in this manuscript were obtained from the GTEx Portal on 10/25/2021. Our study was supported by NIH/NCI under award P50 CA210964, by the Cholangiocarcinoma Foundation, and by PSC Partners Seeking a Cure to L.R.R.. C.I.A. is a Research Scholar of the Cancer Prevention Research Interest of Texas (CPRIT) award RR170048. J.Ra. was partially supported by NHLBI under award K25 HL152006 and by Artificial Intelligence/Machine Learning Consortium to Advance Health Equity and Researcher Diversity (AIM-AHEAD) award OD032581-01S1.

## Author contributions

Y.H., J.B., and C.I.A. conceived and designed the study; Y.H. prepared and curated data; Y.H. and J.B. carried out the analyses and wrote the first draft of the manuscript; R.S. performed multi-omic annotation analysis; **J.Y.R**. assisted the description of results from drug repositioning analysis; C.Z., H.J.C., H.L., S.W.K., **J.Ra**., V.R.S., M.A.C., M.M.H., K.A.M., and L.R.R., C.I.A. contributed to interpretation of the results; T.F., D.E., A.B., S.M.R., A.F., T.H.K., K.N.L., and IPSCSG provided the summary statistics of PSC GWAS; H.J.C. and K.A.S. provided the summary statistics of PBC GWAS; Y.H., J.B., and C.I.A. supervised the study; all authors provided critical feedback and revised the manuscript for important intellectual content.

## Competing interests

The authors declare no competing interests.

## Additional information

[1]Institute for Clinical and Translational Research, Baylor College of Medicine, Houston, TX, USA. [2]Section of Epidemiology and Population Sciences, Department of Medicine, Baylor College of Medicine, Houston, TX, USA. [3]Dan L Duncan Comprehensive Cancer Center, Baylor College of Medicine, Houston, TX, USA. [4]Department of Biostatistics, University of Texas, M.D. Anderson Cancer Center, Houston, TX, USA. [5]Department of Pharmacy, Ochsner Health, New Orleans, LA, USA. [6]Population Health Sciences Institute, Faculty of Medical Sciences, Newcastle University, Newcastle upon Tyne, United Kingdom. [7]David J. Sugarbaker Division of Thoracic Surgery, Michael E. DeBakey Department of Surgery, Baylor College of Medicine, Houston, TX, USA. [8]VA HSR&D, Center for Innovations in Quality, Effectiveness and Safety, Michael E. DeBakey VA Medical Center, Houston, TX, USA. [9]Big Data Scientist Training Enhancement Program (BD-STEP), VA Office of Research and Development, Washington, DC, USA. [10]Department of Medicine, Baylor College of Medicine, Houston, TX, USA. [11]VA Quality Scholars Coordinating Center, IQuESt, Michael E. DeBakey VA Medical Center, Houston, TX, USA. [12]Mayo Clinic Graduate School of Biomedical Sciences, Mayo Clinic, Rochester, MN, USA. [13]Department of Epidemiology, The University of Texas MD Anderson Cancer Center, Houston, TX, USA. [14]Departments of Medicine, Immunology and Medical Sciences, University of Toronto, Toronto, Ontario, Canada. [15]Mount Sinai Hospital, Lunenfeld-Tanenbaum Research Institute and Toronto General Research Institute, Toronto, Ontario, Canada. [16]Norwegian PSC Research Center, Oslo University Hospital Rikshospitalet, Oslo, Norway. [17]Institute of Clinical Molecular Biology, Christian-Albrechts-University of Kiel, Kiel, Germany. [18]Department of Medicine Huddinge, Unit of Gastroenterology and Rheumatology, Karolinska Institutet, Karolinska University Hospital, Stockholm, Sweden. [19]Department of Gastroenterology, Norfolk and Norwich University Hospital, Norwich, United Kingdom. [20]Norwich Medical School, University of East Anglia, Norfolk, United Kingdom. [21]Oslo University Hospital Rikshospitalet and University of Oslo, Oslo, Norway. [22]Division of Gastroenterology and Hepatology, Department of Internal Medicine, Mayo Clinic, Rochester, MN, USA. [23]Division of Cancer Epidemiology and Genetics, National Cancer Institute, Rockville, MD, USA. [27]These authors contributed equally: Younghun Han, Jinyoung Byun. ✉e-mail: Chris.Amos@bcm.edu

## The International PSC Study Group

**Christoph Schramm**[24], **David Shapiro**[25] & **Elizabeth Goode**[26]

[24]1st Department of Medicine, University Medical Center Hamburg-Eppendorf, Hamburg, Germany. [25]4350 La Jolla Village Drive Suite 960, San Diego, CA, USA. [26]Academic Department of Medical Genetics, University of Cambridge, Cambridge, UK. A full list of members and their affiliations appears in the Supplementary Information.

