## [Peer Review File · Nature Communications]

Multi-trait genome-wide analyses identify new susceptibility loci and candidate drugs to primary sclerosing cholangitisREVIEWER COMMENTS

Reviewer #1 (Remarks to the Author):

This manuscript details the results of a genome wide joint analysis to identify novel risk loci for primary sclerosing cholangitis (PSC). The authors leveraged GWAS summary statistics from multiple traits related to PSC, including IBD, lupus, primary biliary cirrhosis (PBC), among others as well as other polygenic traits. To this end the authors identify eight new risk loci not previously described in PSC and replicate using an independent dataset. They go on to show that co-localization with eQTL using data from 49 tissues generated by GTEx, and then go on to perform a pathway analysis to identify potential drug targets from PSC. While the approaches used and the results presented in this manuscript are reveal important new insights into PSC, there are some concerns with the study design and the very limited downstream analyses presented to help determine the impact of these noncoding variants on disease. Moreover, the authors do not discuss the extensive functional genetic work that has been published in some of the regions they report, such as IRF5 and TNFAIP3.

- 1). It is not clear if the data presented in Figure 3A and B are representative of previously reported PSC GWAS. Which regions in these figures represent loci identified when not using this approach as there are many associations that are not annotated in 3B. In addition, it is not clear if the 8 novel regions showed any SNP-PSC association or if they only emerged in the MTAG. It would be expected that these regions would show suggestive association with PSC at some level below genome-wide significance.
- 2). Some of the summary statistic data used for this study were quite old and do not represent the currently identified risk loci. For example, the authors used a lupus dataset from Bentham et al. 2015 while also citing Langefeld et al. 2017 which reports many more risk loci in Europeans. Moreover, with the extreme difference in the number of subjects and loci associated between the traits used in this study could lead to bias in the findings that it is not clear was accounted for in their model.
- 3). The co-localization approaches used in this study only looked for relationships between the associated variants and data from GTEx. While this resource provides a lot of diverse tissues to be tested, it does not look at the immune system other than whole blood. Since PSC is an autoimmune disease that impacts the bile duct and is associated with other related diseases, such as PBC and IBD, the authors should have used other resources that look at more immune cell subsets such as those arrayed in FUMA. Many of the non-coding variants likely impact enhancers that are known to act in cell type and context specific manner. It is not clear how the current approach increases our understanding of how PSC risk alleles impact the immune system with the current approach. The authors could also use many of the other publicly available data to help identify regulation outside the nearest gene to the association signal, such as promoter capture Hi-C.
- 4). There have been many publications regarding some of the risk loci the authors describe in this work, such as IRF5 (Kottyan et al. 2015 Hum Mol Genet) and TNFAIP3 (Wang et al. 2016 Genes Immun), that the authors ignore. Since there are no functional studies presented here, why were the variants not considered that these studies suggest.
- 5). The pathway analysis used in this study only seemed to use a limited set of genes for each region associated with PSC often only including one for each region/effect they have identified as associated with PSC. It is likely there are many genes outside the index (nearest) gene to the association signal could be regulated by the risk haplotype, but the authors do not consider this in their work.
- 6). It would be helpful to indicate the number of PSC cases present in the replication study.

Reviewer #2 (Remarks to the Author):

Han et al., performed multi-trait joint analyses to genome-wide association summary statistics of PSC and other traits to identify new loci and performed network-based analysis for identifying candidate pharmacological agents.

Major Comments-

None

Minor Comments-

1. Why was MHC excluded from the LDSR analysis?
2. Smoking cessation was positively correlated with PSC. How do the authors explain this finding? Especially the current smoking was protective as compared to past smoking. Is it just an artefact of the data? It should be stated with caution so as not to be misinterpreted.
3. Was the data for duration and number of packs smoked per year available?
4. Was sex (being male) used as a covariate for smoking?
5. What was the ancestry of the PSC GWAS samples?

Re: Response to comments on NCOMMS-22-20861-T. “Multi-trait genome-wide joint analyses reveal new susceptibility loci and predict network-based proximity of drugs to primary sclerosing cholangitis” submitted by Younghun Han et al. to Nature Communications for publication.

REVIEWER COMMENTS

Reviewer #1 (Remarks to the Author):

This manuscript details the results of a genome wide joint analysis to identify novel risk loci for primary sclerosing cholangitis (PSC). The authors leveraged GWAS summary statistics from multiple traits related to PSC, including IBD, lupus, primary biliary cirrhosis (PBC), among others as well as other polygenic traits. To this end the authors identify eight new risk loci not previously described in PSC and replicate using an independent dataset. They go on to show that co-localization with eQTL using data from 49 tissues generated by GTEx, and then go on to perform a pathway analysis to identify potential drug targets from PSC. While the approaches used and the results presented in this manuscript are reveal important new insights into PSC, there are some concerns with the study design and the very limited downstream analyses presented to help determine the impact of these noncoding variants on disease. Moreover, the authors do not discuss the extensive functional genetic work that has been published in some of the regions they report, such as IRF5 and TNFAIP3.

1). *It is not clear if the data presented in Figure 3A and B are representative of previously reported PSC GWAS. Which regions in these figures represent loci identified when not using this approach as there are many associations that are not annotated in 3B. In addition, it is not clear if the 8 novel regions showed any SNP-PSC association or if they only emerged in the MTAG. It would be expected that these regions would show suggestive association with PSC at some level below genome-wide significance.*

Response: We apologize for the confusion. The summary statistics used for Fig 3a and 3b were from previously published single trait GWAS (Ji et al. 2017 NG) and PSC-specific GWAS combined with five immune-mediated disorders using Multi-Trait Analysis of GWAS (MTAG), respectively. We revised the sentence “(a) and (c) are plotted using the PSC single-trait GWAS (Ji et al., 2017, PMID:279922413; GWAS_PSC). (b) and (d) are plotted using the MTAG-identified PSC-specific GWAS combined with five immune-mediated disorders (MTAG_PSC).” in the legend of Fig 3.

Instead of newly MTAG-identified gene names in Fig 3b, we updated all genome-wide significant loci with cytoband in Fig 3a and 3b. MTAG-identified seven novel associations showing a similar magnitude of effect sizes (Odds Ratios, OR) compared to previously reported results from Ji et al. (PMID: 27992413)¹. The last column in Table 2 showed the results from the previous GWAS of Ji et al. for these seven new loci that have not been previously reported and one new independent lead variant in the previously reported locus. These novel genetic signals demonstrated suggestive significant associations with PSC but do not reach at the genome-wide significance level of $P < 5 \times 10^{-8}$.

2). *Some of the summary statistic data used for this study were quite old and do not represent the currently identified risk loci. For example, the authors used a lupus dataset from Bentham et al. 2015 while also citing Langefeld et al. 2017 which reports many more risk loci in Europeans. Moreover, with the extreme difference in the number of subjects and loci associated between the traits used in this study could lead to bias in the findings that it is not clear was accounted for in their model.*

Response: We thank the reviewer for great suggestion to improve our manuscript. We obtained lupus GWAS summary statistics from GWAS Catalog (Langefeld et al. 2017, GCST007400)² and examined the summary statistics for further analyses. We learned that there is discordance between sample sizes from GWAS catalog and those from downloaded summary statistics. While 6,748 cases and 11,516 controls were reported in GWAS

¹ Ji SG, Juran BD, Mucha S, Folseraas T, Jostins L, Melum E, et al. Genome-wide association study of primary sclerosing cholangitis identifies new risk loci and quantifies the genetic relationship with inflammatory bowel disease. Nat Genet. 2017;49(2):269-73. Epub 2016/12/20. doi: 10.1038/ng.3745. PubMed PMID: 27992413; PMCID: PMC5540332.

² Langefeld CD, Ainsworth HC, Cunninghame Graham DS, Kelly JA, Comeau ME, Marion MC, et al. Transancestral mapping and genetic load in systemic lupus erythematosus. Nat Commun. 2017;8:16021. Epub 2017/07/18. doi: 10.1038/ncomms16021. PubMed PMID: 28714469; PMCID: PMC5520018.

Catalog, in actual summary statistics obtained from GWAS catalog, 5,506 cases and 11,323 controls on chromosome 1 to 8 and 4,146 cases and 11,280 controls on chromosome 9 to 22 are available for further analyses. While the number of SNPs overlapping between Bentham et al. (2015) and HapMap 3 are approximately 1.2M, those between Langefeld et al. (2017) and HapMap 3 are approximately 0.57M. Please see the table below for the final number of SNPs among summary-level data of Lupus.

Study	N. SNPs
Lupus (Bentham, 2015)	1,171,324
Lupus (Langefeld, 2017) with QC1	569,936
Lupus (Langefeld, 2017) with QC2	1,186,549
HapMap 3	1,217,311

The summary statistics of Langefeld et al. (2017) is the original output from “SNPTEST” (the number of SNPs =88,534,480). So, we applied two different quality controls of QC1 (MAF= 0.001 and INFO = 0.5) and QC2 (MAF = 0.001 and INFO = 0.3). 6,416,007 and 20,762,284 SNPs were remained for further analysis after QC1 and QC2, respectively. We then compared the results from cross-trait genetic correlation analyses using LD score regression. We tabulated the results from pairwise genetic correlation between PSC and Lupus.

Comparison of results from pairwise genetic correlation between Lupus and PSC using LDSR.

Study	rg	se	z	p	h2_obs	h2_obs_se	h2_int	h2_int_se
Lupus (Bentham, 2015)	0.20	0.10	2.03	4.19×10^{-2}	0.42	0.07	1.10	0.01
Lupus (Langefeld, 2017) with QC1	0.12	0.11	0.28	0.28	0.52	0.12	1.12	0.03
Lupus (Langefeld, 2017) with QC2	-0.01	0.10	-0.12	0.90	0.49	0.10	1.03	0.02

In addition to discordant sample size, the estimated genetic correlation between PSC and lupus (Langefeld et al., 2017) is 0.12, which does not meet our criteria ($\text{abs}(\text{genetic correlation}) > 0.20$) for selecting autoimmune-related traits. Thus, summary statistics for lupus from Bentham et al. 2015 is more suitable for multi-trait joint analyses in the current study design.

We performed the MTAG to compare the results using Lupus (Bentham, 2015) and Lupus (Langefeld, 2017) with QC1 (MAF= 0.001 and INFO = 0.5) even though Lupus (Langefeld, 2017) did not meet our criteria of $\text{abs}(\text{genetic correlation}) > 0.20$. We compared the P-values between the results of MTAG with lupus_Bentham and lupus_Langefeld for 19 SNPs (7 new loci, 1 new lead variant in the previously reported locus, and 11 previously reported variants). As we expected, the power of MTAG with lupus_Bentham (Bentham et al. 2015) is slightly better than those with lupus_Langefeld (Langefeld et al. 2017). Also, we conducted the sensitivity analysis after excluding lupus_Langefeld for 19 SNPs. The table below demonstrates the comparison of MTAG results modeled in PSC, CD, UC, IBD, and PBC with lupus_Bentham and lupus_Langefeld, respectively.

SNP	POS	Cytoband	Gene	MTAG with lupus Bentham			MTAG with lupus Langefeld			MTAG sensitivity (excluding lupus Langefeld)		
				Beta	SE	P	Beta	SE	P	Beta	SE	P
rs3748816	2526746	1p36.32	MMEL1	-0.068	0.011	1.22E-09	-0.068	0.011	6.57E-10	-0.067	0.012	5.23E-09
rs7608697	61204641	2p16.1	PUS10	0.064	0.011	3.11E-09	0.050	0.011	2.35E-06	0.065	0.011	3.92E-09
rs7426056	204612058	2q33.2	CD28 CTLA4	-0.076	0.012	5.96E-10	-0.073	0.012	1.05E-09	-0.076	0.012	1.53E-09
rs6787808	49079105	3p21.31	QRICH1	0.074	0.012	1.20E-09	0.061	0.012	3.53E-07	0.078	0.012	3.75E-10
rs3197999	49721532	3p21.31	MST1	0.101	0.012	3.85E-18	0.087	0.011	2.22E-14	0.107	0.012	2.84E-19
rs228614	103578637	4q24	MANBA	-0.064	0.011	1.71E-09	-0.053	0.010	3.40E-07	-0.066	0.011	1.03E-09
rs13140464	123499745	4q27	IL2 IL21	-0.091	0.014	2.42E-10	-0.094	0.014	1.79E-11	-0.096	0.015	6.41E-11
rs12198665	39240796	6p21.2	KCNK5 KCNK17	-0.064	0.012	3.35E-08	-0.056	0.011	5.29E-07	-0.068	0.012	1.01E-08
rs17780429	138222588	6q23.3	TNFAIP3 LINC02528	-0.094	0.015	2.24E-10	-0.094	0.014	6.29E-11	-0.089	0.015	3.78E-09
rs3757387	128576086	7q32.1	KCP IRF5	0.081	0.011	2.19E-14	0.092	0.010	4.98E-19	0.060	0.011	3.56E-08
rs4147359	6108439	10p15.1	IL2RA RBM17	0.083	0.011	5.07E-14	0.092	0.011	1.47E-17	0.088	0.011	5.95E-15
rs7911680	101293468	10q24.2	NKX2-3	-0.060	0.011	1.33E-08	-0.055	0.010	1.10E-07	-0.066	0.011	1.25E-09
rs663743	64107735	11q13.1	CCDC88B	-0.065	0.011	6.27E-09	-0.061	0.011	3.61E-08	-0.066	0.012	1.08E-08
rs3184504	111884608	12q24.12	SH2B3	-0.085	0.011	1.07E-15	-0.072	0.010	2.72E-12	-0.081	0.011	8.57E-14
rs725613	11169683	16p13.13	CLEC16A	-0.081	0.011	3.50E-13	-0.075	0.011	4.63E-12	-0.078	0.011	5.79E-12
rs79390277	68942590	16q22.1	TANGO6	0.137	0.024	1.69E-08	0.119	0.024	4.86E-07	0.138	0.025	2.73E-08
rs1788097	67543688	18q22.2	CD226	0.061	0.011	6.82E-09	0.055	0.010	1.16E-07	0.058	0.011	6.72E-08
rs60652743	47205707	19q13.32	PRKD2	-0.088	0.014	1.11E-09	-0.087	0.014	6.47E-10	-0.090	0.015	9.54E-10
rs2836883	40466744	21q22.2	LINC01700 PSMG1	-0.115	0.012	1.56E-21	-0.102	0.012	6.13E-18	-0.121	0.012	8.51E-23

Since we exploited the existing summary statistics as they have been presented in the literature, we applied appropriate quality control filters to take into account the difference in the number of subjects and loci associated between the traits used in this study. The rationale of MTAG is that the estimated effect for each trait can be improved by appropriately incorporating information contained in the GWAS estimates for other traits tested together, when GWAS estimates from different traits are correlated. We first surveyed the estimated SNP-heritability and estimated pairwise genetic correlation showing moderate proportion of phenotypic variance explained by all common SNPs ($h^2 > 0.2$) and moderate to high level of polygenic overlap between polygenic traits tested ($|\text{rg}| > 0.2$).

3). *The co-localization approaches used in this study only looked for relationships between the associated variants and data from GTEx. While this resource provides a lot of diverse tissues to be tested, it does not look at the immune system other than whole blood. Since PSC is an autoimmune disease that impacts the bile duct and is associated with other related diseases, such as PBC and IBD, the authors should have used other resources that look at more immune cell subsets such as those arrogated in FUMA. Many of the non-coding variants likely impact enhancers that are known to act in cell type and context specific manor. It is not clear how the current approach increases our understanding of how PSC risk alleles impact the immune system with the current approach. The authors could also use many of the other publicly available data to help identify regulation outside the nearest gene to the association signal, such as promoter capture Hi-C.*

Response: Following the reviewer's suggestion, we explored immune cell subsets that potentially influence PSC risk through FUMA³ and InnateDB⁴. We found 48 genes by eQTL associated with the expression of 14 immune cell types using FUMA and 5 immune-related genes including *IRF5* and *SMO* (7q32.1) and *HAS3*, *SNTB2*, and

³ Watanabe K, Taskesen E, van Bochoven A, Posthuma D. Functional mapping and annotation of genetic associations with FUMA. Nat Commun. 2017;8(1):1826. Epub 2017/12/01. doi: 10.1038/s41467-017-01261-5. PubMed PMID: 29184056; PMCID: PMC5705698.

⁴ Breuer K, Foroushani AK, Laird MR, Chen C, Sribnaia A, Lo R, et al. InnateDB: systems biology of innate immunity and beyond--recent updates and continuing curation. Nucleic Acids Res. 2013;41(Database issue):D1228-33. Epub 2012/11/28. doi: 10.1093/nar/gks1147. PubMed PMID: 23180781; PMCID: PMC3531080.

VPS4A (16q22.1), within new loci that have not previously reported using InnateDB. We reported the findings in **Supplementary Table 6 and 7** and revised the manuscript according to the updated results in the section of “Results”.

We also examined how the non-coding variants associated with MTAG-identified PSC susceptibility act in cell type and functional context using integrated variant functional annotation approach⁵ through the Functional Annotation of Variants-Online Resource (FAVOR)⁶ platform and the Multi-dimensional Annotation Class Integrative Estimator^{7,8} (MACIE). We identified 168 non-coding genes within 20 MTAG-identified genomic risk loci. Among them, we observed 14 more likely deleterious genes with CADD score ≥ 12.37 , with the threshold recommended by Kircher et al. (Nature Genetics, 2014)⁹ and 8 and 6 genes on promoter and permissive enhancer sites, respectively. Of the SNPs investigated with MACIE, we find 80 variants with a regulatory class prediction greater than 95%. That is, these variants are highly likely to tangibly affect the behavior of certain gene expressions, most often nearby genes. We find 4 variants with a conserved class prediction greater than 95%, and three of these variants also possess a regulatory prediction greater than 95%. That is, the 4 variants are highly likely to belong to the class of evolutionarily conserved variants that are found in many living beings. The full predictions for each SNP can be found in **Supplementary Table 10**.

Out of 406 prioritized genes, 48 genes (12%) were detected by eQTL associated with the expression of 14 immune cell types. In the chromatin interaction mapping, 278 genes (69%) are mapped to the regions interacting with the promoter of the listed gene and of which 90 genes (32%) were found in the liver tissue in which the chromatin interaction is observed (**Supplementary Table 6**). Either chromatin interactions or eQTLs within PSC risk loci (**Supplementary Table 5**) were shown on chromosome 2, 3, 4, 6, 7, 11, 16, 19, and 21, respectively (**Supplementary Figures 3**). 158 genes were mapped by both eQTLs and chromatin interactions including *IRF5* and *TNPO3* genes (in red on the **Supplementary Figure 3(e)**) on the 7q32.1. The new findings are revised in the section of “Results”.

While PSC GWAS have identified a few risk-associated variants, the function and identity of the causal variants are not fully explored. To address (1) alleles within less-well-established non-coding regions, (2) the impact of each variant in the immune system in the current study, we highlight several *in silico* functional approaches to map and prioritize the variants identified. We demonstrated that 48 coding genes out of 406 prioritized genes are observed in eQTL associated with the expression of 14 immune cell types. Furthermore, we exploited immune-related gene database for deciphering how PSC risk-associated alleles may alter immune networks through Gene Ontology pathway. We also utilized the integrative functional annotations platform to functionally characterize the prioritized genes including both coding and non-coding genes, which provide numerous information of variant and indel functional annotations. For example, DNase and H3k27ac can provide how PSC risk-associated alleles localize within open chromatin. Since there is no medication proven to treat PSC, we predicted many potential agents at the relative proximity capturing the statistically significant relationship between potential drug and putative disease-associated proteins using 32 MTAG-identified genes available in NCBI reference gene catalog. We further carried out gene mapping to drug database with the broad range of genes prioritized by position, eQTL, and chromatin interaction mapping. These analytical pipelines, which utilize activity maps of non-coding regions help us pinpoint their role in specific cell types. These findings can provide better functional insight into the genetic etiology of PSC susceptibility and improve our understanding of how PSC risk-associated variants alter immune system.

⁵ Li Z, Li X, Zhou H, Gaynor SM, Selvaraj MS, Arapoglou T, et al. A framework for detecting noncoding rare variant associations of large-scale whole-genome sequencing studies. bioRxiv. 2021:2021.11.05.467531. doi: 10.1101/2021.11.05.467531.

⁶ Zhou H, Arapoglou T, Li X, Li Z, Zheng X, Moore J, et al. FAVOR: Functional Annotation of Variants Online Resource and Annotator for Variation across the Human Genome. bioRxiv. 2022:2022.08.28.505582. doi: 10.1101/2022.08.28.505582.

⁷ Sun, R. *et al.* Integration of multiomic annotation data to prioritize and characterize inflammation and immune-related risk variants in squamous cell lung cancer. *Genet Epidemiol* **45**, 99-114 (2021).

⁸ Li, X. *et al.* A multi-dimensional integrative scoring framework for predicting functional variants in the human genome. *Am J Hum Genet* **109**, 446-456 (2022).

⁹ Kircher M, Witten DM, Jain P, O’Roak BJ, Cooper GM, Shendure J. A general framework for estimating the relative pathogenicity of human genetic variants. *Nat Genet.* 2014;46(3):310-5. Epub 2014/02/04. doi: 10.1038/ng.2892. PubMed PMID: 24487276; PMCID: PMC3992975.

4). There have been many publications regarding some of the risk loci the authors describe in this work, such as *IRF5* (Kottyan et al. 2015 *Hum Mol Genet*) and *TNFAIP3* (Wang et al. 2016 *Genes Immun*), that the authors ignore. Since there are no functional studies presented here, why were the variants not considered that these studies suggest.

Response: We thank the reviewer for the comment. Wang et al. demonstrated that the region of *TNFAIP3* significantly associated with autoimmune disease such as systemic lupus erythematosus has significant regulatory functions¹⁰. To examine functional characterization of causal genetic variants, they employed transcription activation-like effector nuclease-mediated genome-editing technology. In the present study, we implemented MACIE^{7,8} to examine functional characterization. The MTAG-identified PSC-specific risk variant, rs17780429 in *TNFAIP3* seems to have a regulatory class prediction greater than 99%, implying that rs17780429 in *TNFAIP3* is highly likely to affect the behavior of certain gene expression.

Kottyan et al. surveyed variants in or near *IRF5* associated with in two immunotherapies and seven autoimmune diseases to identify the causal variants and constructed a genetic model explaining the genetic association of variants in *IRF5-TNPO3* using Bayesian and frequentist approaches¹¹. We checked *IRF5-TNPO3* associations reported in Kottyan et al.'s work. We observed genome-wide significant MTAG-identified PSC risk associations including rs4728142 ($P_{\text{MTAG_PSC}} = 4.00 \times 10^{-13}$) in the *IRF5* promoter region, rs2004640 ($P_{\text{MTAG_PSC}} = 2.45 \times 10^{-9}$), rs3807306 ($P_{\text{MTAG_PSC}} = 4.39 \times 10^{-12}$), rs10954213 ($P_{\text{MTAG_PSC}} = 4.21 \times 10^{-9}$) in *IRF5*, and rs10488631 ($P_{\text{MTAG_PSC}} = 1.31 \times 10^{-8}$), rs12534421 ($P_{\text{MTAG_PSC}} = 1.33 \times 10^{-8}$) in *TNPO3* in European ancestry population while these associations in a single-disease GWAS of PSC¹ did not show the genome-wide significant signals. To nominate the causal variants from each locus for further functional analysis, we implemented fine-mapping of MTAG-identified loci using FINEMAP¹² package and surveyed credible sets of plausible causal variants based on posterior inclusion probability (PIP). Based on the single-SNP PIP within each locus, we identified 32 variants falling into the 95% credible set across 8 MTAG-identified GWAS loci. From the MTAG-identified PSC risk locus in *IRF5* promoter region at 7q32.1 rs3757387, one of five causal variants found within 95% credible set in both Kottyan's and our study were predicted to have regulatory potential with marginal probability of 0.57. The results are reported in **Supplementary Table 10 and 11**.

Leveraging *in silico* functional approaches, we prioritized the candidate genes by eQTL and chromatin interaction mapping underlying the MTAG-identified PSC-specific associations using FUMA and functionally characterized the 329 independent significant variants within 20 MTAG-identified genomic risk loci using FAVOR platform. We revised the manuscript according to the updated results in the section of "Results".

5). The pathway analysis used in this study only seemed to use a limited set of genes for each region associated with PSC often only including one for each region/effect they have identified as associated with PSC. It is likely there are many genes outside the index (nearest) gene to the association signal could be regulated by the risk haplotype, but the authors do not consider this in their work.

Response: We thank the reviewer for pointing this out. In the earlier version, we examined functional enrichment of protein-protein interaction networks and functional annotations amongst MTAG-identified candidate genes using the STRING Database. We performed additional functional enrichment analysis such as GO and KEGG pathways using 406 genes mapped from position mapping, eQTL mapping, and chromatin interaction mapping using the Database for Annotation, Visualization, and Integrated Discovery (DAVID) Bioinformatics

¹⁰ Wang, S., Wen, F., Tessneer, K.L. & Gaffney, P.M. TALEN-mediated enhancer knockout influences *TNFAIP3* gene expression and mimics a molecular phenotype associated with systemic lupus erythematosus. *Genes Immun* **17**, 165-70 (2016).

¹¹ Kottyan, L.C. et al. The *IRF5-TNPO3* association with systemic lupus erythematosus has two components that other autoimmune disorders variably share. *Hum Mol Genet* **24**, 582-96 (2015).

¹² Benner C, Spencer CC, Havulinna AS, Salomaa V, Ripatti S, Pirinen M. FINEMAP: efficient variable selection using summary data from genome-wide association studies. *Bioinformatics*. 2016;32(10):1493-501. Epub 2016/01/17. doi: 10.1093/bioinformatics/btw018. PubMed PMID: 26773131; PMCID: PMC4866522.

Resources^{13,14} (v 6.8). The best practice for selecting SNPs and the genes they affect is not yet well resolved, but because of linkage disequilibrium, selecting all the SNPs in a region associated with a disease would lead to overweighting of regions that include large linkage disequilibrium blocks. Therefore, we further annotated SNPs prior to selection of them for pathway analysis. We updated our findings in the section of “Results” as follows:

“We selected 406 prioritized genes to detect relevant groups of related genes involved in the regulation of specific biological pathways. Using STRING Protein-Protein Interaction (PPI) networks¹, these candidate genes are highly enriched for protein-protein interactions ($P < 1.00 \times 10^{-16}$), with enrichment at false discovery rate (FDR) < 0.05 of the following pathways: Immune receptor activity (FDR= 3.84×10^{-2}), beta-2-microglobulin binding (1.10×10^{-2}), Cytokine-mediated signaling pathway (1.58×10^{-13}), Interferon-gamma-mediated signaling pathway (1.13×10^{-11}), T cell receptor signaling pathway (2.21×10^{-11}), Immune response-activating cell surface receptor signaling pathway (2.65×10^{-9}), interleukin-7-mediated signaling pathway (9.21×10^{-9}), TNFR2 non-canonical NF- κ B pathway (7.90×10^{-3}), Th17 cell differentiation (2.63×10^{-6}), and Th1 and Th2 cell differentiation (1.94×10^{-5}) (**Supplementary Table 13, Supplementary Figure 5**). For comparison, we implemented enrichment analysis using DAVID Bioinformatics Resources (v 6.8) on the same candidate 406 genes. We observed T cell receptor signaling pathway (FDR= 5.82×10^{-7}), antigen processing and presentation (8.18×10^{-15}), Immunoglobulin production involved in immunoglobulin mediated immune response (6.30×10^{-14}), Cytokine Signaling in Immune system (3.48×10^{-5}), Interferon Signaling (2.62×10^{-9}), and Interferon alpha/beta signaling (6.60×10^{-4}) (**Supplementary Table 14**).

In addition, we scrutinized the PPI network associated with each gene prioritized from newly MTAG-identified loci. We found four genes (*MANBA*, *IRF5*, and *NKX2-3*) to be highly enriched for PPI at FDR < 0.05 . The prioritized genes, *MANBA*, *IRF5*, and *NKX2-3* reported PPI P-values of 5.16×10^{-14} , 1.00×10^{-16} , and 1.13×10^{-9} , respectively. We observed B and T cell receptor, Chemokine, C-type lectin receptor, Cytosolic DNA-sensing, HIF-1, IL-17, JAK-STAT, MAPK, Metabolic, NF-kappa B, NOD-like receptor, PD-L1 expression and PD-1 checkpoint in cancer, RIG-I-like receptor, th1-th2 cell differentiation, th17 cell differentiation, Thyroid hormone, TNF, and Toll-like receptor signaling pathways in the KEGG pathways at FDR < 0.05 using STRING PPI networks (**Supplementary Table 15, Supplementary Figure 6**).”

The results from STRING PPI and DAVID functional enrichment analyses on 406 prioritized genes at FDR < 0.05 were added into “**Supplementary Table 13 and 14**” and the PPI networks within 416 prioritized multiple proteins were displayed in **Supplementary Figure 5**.

6). *It would be helpful to indicate the number of PSC cases present in the replication study.*

Response: PSC GWAS from FinnGen consists of 952 cases and 231,644 controls. A new column with the number of cases was added in the **Supplementary Table 2**.

¹³ Huang da, W., Sherman, B.T. & Lempicki, R.A. Systematic and integrative analysis of large gene lists using DAVID bioinformatics resources. *Nat Protoc* **4**, 44-57 (2009).

¹⁴ Sherman, B.T. *et al.* DAVID: a web server for functional enrichment analysis and functional annotation of gene lists (2021 update). *Nucleic Acids Res* (2022).

Reviewer #2 (Remarks to the Author):

Han et al., performed multi-trait joint analyses to genome-wide association summary statistics of PSC and other traits to identify new loci and performed network-based analysis for identifying candidate pharmacological agents.

Major Comments-

None

Minor Comments-

1. *Why was MHC excluded from the LDSR analysis?*

Response: We thank the reviewer for great suggestions to improve our manuscript. The SNPs within the major histocompatibility complex (MHC) region (Chr6:25-34Mb) were excluded from the LDSR analysis since these SNPs show the complex and unusual LD pattern or extreme effect sizes^{15,16}. It is recommended to remove the MHC region in the LDSR analysis by the software developers.

2. *Smoking cessation was positively correlated with PSC. How do the authors explain this finding? Especially the current smoking was protective as compared to past smoking. Is it just an artefact of the data? It should be stated with caution so as not to be misinterpreted.*

Response: We thank the reviewer for raising the potential concerns. We downloaded four summary statistics related to smoking behaviors (AgeSmk, GCST007458; SmkInit, GCST007474; CigDay, GCST007459; SmkCes, GCST007460) from GWAS Catalog (Liu et al., 2019) and performed LDSR analysis modeled with the consistent analytical setup. Smoking cessation (SmkCes) modeled in former smokers versus current smokers showed protective (negative) genetic correlation with PSC ($rg = -0.11$; $P = 9.99 \times 10^{-2}$) although it failed to meet the nominal significance level of 0.05. Smoking initiation (SmkInit) modeled in never smokers versus ever smokers showed a significant negative genetic correlation with PSC ($rg = -0.20$; $P = 2.05 \times 10^{-6}$). The findings from LDSR were consistent with previously published epidemiological studies. We revised the sentences in the sections of 'Results' and 'Discussions' as follows:

“Among traits related to smoking behaviors in this study, smoking status modeled in previous smokers versus current smokers showed a strong negative genetic correlation with PSC susceptibility ($rg = -0.27$; $P = 9.17 \times 10^{-10}$) while smoking initiation, which is a binary phenotype indicating whether an individual had ever smoked regularly (i.e., never smokers versus ever smokers), reported a significant negative genetic correlation with PSC ($rg = -0.20$; $P = 2.05 \times 10^{-6}$).”

“Several epidemiological studies have reported inverse associations between smoking and PSC risk. Our study found a strongly protective genetic correlation between the genomic architecture of smoking status modeled in former smokers versus current smokers and that of PSC, suggesting that the genetic contribution of current smoking is associated with a decreased risk of PSC compared to that of former smoking. Although it failed to meet the Bonferroni-corrected significance level of 3.73×10^{-4} , the smoking cessation trait modeled in former smokers versus current smokers showed a consistent association with PSC implying that the genetic contribution of current smoking is associated with a decreased risk of PSC compared to that of former smoking. The smoking initiation trait modeled in never smokers versus ever smokers showed a significant negative association with PSC suggesting that PSC risk among current and former smokers is significantly lower than that among never smokers. Smoking promotes chronic epithelial and tissue injury through chronic airway inflammation and the most common causes of chronic inflammation include immune-mediated disorders which could potentially contribute

¹⁵ Byun J, Han Y, Walsh KM, Park AS, Bondy ML, Amos CI. Shared genomic architecture between COVID-19 severity and numerous clinical and physiologic parameters revealed by LD score regression analysis. *Sci Rep.* 2022;12(1):1891. Epub 2022/02/05. doi: 10.1038/s41598-022-05832-5. PubMed PMID: 35115602.

¹⁶ Zheng J, Erzurumluoglu AM, Elsworth BL, Kemp JP, Howe L, Haycock PC, et al. LD Hub: a centralized database and web interface to perform LD score regression that maximizes the potential of summary level GWAS data for SNP heritability and genetic correlation analysis. *Bioinformatics.* 2017;33(2):272-9. Epub 2016/11/03. doi: 10.1093/bioinformatics/btw613. PubMed PMID: 27663502; PMCID: PMC5542030.

to PSC development. Therefore, the shared association of PSC with smoking behaviors makes disentangling such effects challenging.”

We updated the results of genetic correlation between PSC and polygenic traits associated with smoking behaviors and added how to code the smoking status for consistency in the **Supplementary Table 3**.

3. *Was the data for duration and number of packs smoked per year available?*

Response: Yes, the data for ‘Heaviness of smoking was measured with cigarettes per day (CigDay)’ (GWAS Catalog: GCST007459) and ‘Age of initiation of regular smoking (AgeSmk)’ (GCST007458) from Liu et al. (PMID:30643251) are available in GWAS Catalog. The results were reported under a category ‘Smoking Behaviors’ in **Supplementary Table 3**.

4. *Was sex (being male) used as a covariate for smoking?*

Response: Yes, sex was used as a covariate for smoking-related traits¹⁷ (Liu et al., 2019; PMID:30643251).

5. *What was the ancestry of the PSC GWAS samples?*

Response: Summary statistics from PSC GWAS samples of European ancestry were retained in this study since PSC is more common among Northern European populations than among other populations.

¹⁷ Liu M, Jiang Y, Wedow R, Li Y, Brazel DM, Chen F, et al. Association studies of up to 1.2 million individuals yield new insights into the genetic etiology of tobacco and alcohol use. *Nat Genet.* 2019;51(2):237-44. Epub 2019/01/16. doi: 10.1038/s41588-018-0307-5. PubMed PMID: 30643251; PMCID: PMC6358542.

REVIEWERS' COMMENTS

Reviewer #1 (Remarks to the Author):

No further comments.